# Learning auditory discriminations from observation is efficient but less robust than learning from experience

Gagan Narula [1,2], Joshua A. Herbst[1,2], Joerg Rychen[1] & Richard H.R. Hahnloser[1,2]

Social learning enables complex societies. However, it is largely unknown how insights obtained from observation compare with insights gained from trial-and-error, in particular in terms of their robustness. Here, we use aversive reinforcement to train "experimenter" zebra finches to discriminate between auditory stimuli in the presence of an "observer" finch. We show that experimenters are slow to successfully discriminate the stimuli, but immediately generalize their ability to a new set of similar stimuli. By contrast, observers subjected to the same task are able to discriminate the initial stimulus set, but require more time for successful generalization. Drawing on concepts from machine learning, we suggest that observer learning has evolved to rapidly absorb sensory statistics without pressure to minimize neural resources, whereas learning from experience is endowed with a form of regularization that enables robust inference.

[1] Institute of Neuroinformatics, University of Zurich and ETH Zurich, Winterthurerstrasse 190, 8057 Zurich, Switzerland. [2] Neuroscience Center Zurich, University of Zurich and ETH Zurich, Winterthurerstrasse 190, 8057 Zurich, Switzerland. Correspondence and requests for materials should be addressed to R.H.R.H. (email: rich@ini.ethz.ch)

Humans and animals have the remarkable ability to generalize their acquired knowledge to new examples and situations[1–4]. For example, they can learn to discriminate threatening from harmless stimuli and they can generalize this knowledge to new instances of a threat. They are also capable of learning from few examples[5,6], presumably because brains have evolved under the pressure of fatal consequences when threats are not immediately recognized. Two ethologically relevant learning metrics are thus the acquisition time and the transferability of acquired information. Which forms of learning focus more on the former and which more on the latter of these metrics?

We propose a comparative approach towards disentangling rapid learning from robust generalization, exploiting the fact that many animals are not only capable of learning from aversive or appetitive cues through trial-and-error type processes[7,8], but also from observing cues produced by conspecifics and other animals involved in learning or doing the same task[9–11]. In what way does sensory discrimination learning depend on whether the learning cue is experienced or observed, keeping all other parameters fixed?

We study cue dependence of an auditory stimulus discrimination task involving pairs of zebra finches[12–15].

Using aversive air-puffs, we trained one of the two birds in a pair to behaviorally discriminate short from long renditions of a zebra finch song syllable (Go-NoGo avoidance conditioning, Fig. 1a; spectrograms of stimuli in Fig. 1b, durations in Fig. 1c). We refer to these birds as "experimenters". Simultaneously, we allowed a paired zebra finch to observe the entire training phase of the experimenter, including the acoustic stimuli and the experimenter's actions. These latter birds are referred to as "observers"; they could engage in unrestricted visual and auditory interactions with experimenters, but did not perform the task until after experimenters completed their training phase.

The acoustic stimuli in such experiments are fully predictive of whether an air-puff is imminent or not. Experimenters reveal their ability to discriminate the stimuli by escaping from the perch before they get struck by the air-puff[15]. We refer to this form of learning as experience learning because birds learn to discriminate based on experience of the air-puffs. By contrast, observers could not learn from air-puff experiences, but they could learn from observing the air-puffs' direct and indirect effects on experimenters' behaviors. We refer to this form of learning as observation learning (which is not meant to imply that observers learn by imitating the actions of experimenters, which is commonly known as 'observational learning').

We expected that observers would be able to demonstrate their learned discrimination ability in a separate testing phase in which they were exposed to air-puffs. Here, we investigate the performance tradeoffs between experience learning and observation learning using two important metrics of learning: learning speed and generalization performance. We find that observation learning is fast but less robust than experience learning.

## Results

**Observation learning induces rapid auditory discrimination.** During a pre-training phase, experimenters (EXP) were accustomed to air-puffs that followed one of two auditory stimuli of different duration. Then a training phase followed, during which we exposed EXP to the full training set of 10 auditory stimuli (Fig. 2a left panel and Supplementary methods). Gradually, EXP learned to escape from the perch more often in puffed trials; their escape probabilities (cumulated from trial onset to trial end) became larger on puffed trials than on unpuffed trials (Fig. 2b left and right). We quantified the birds' ability to discriminate stimulus class by the difference in (cumulative) escape probabilities (dPesc) between puffed and unpuffed trials (Fig. 2b, c, d, e). EXP

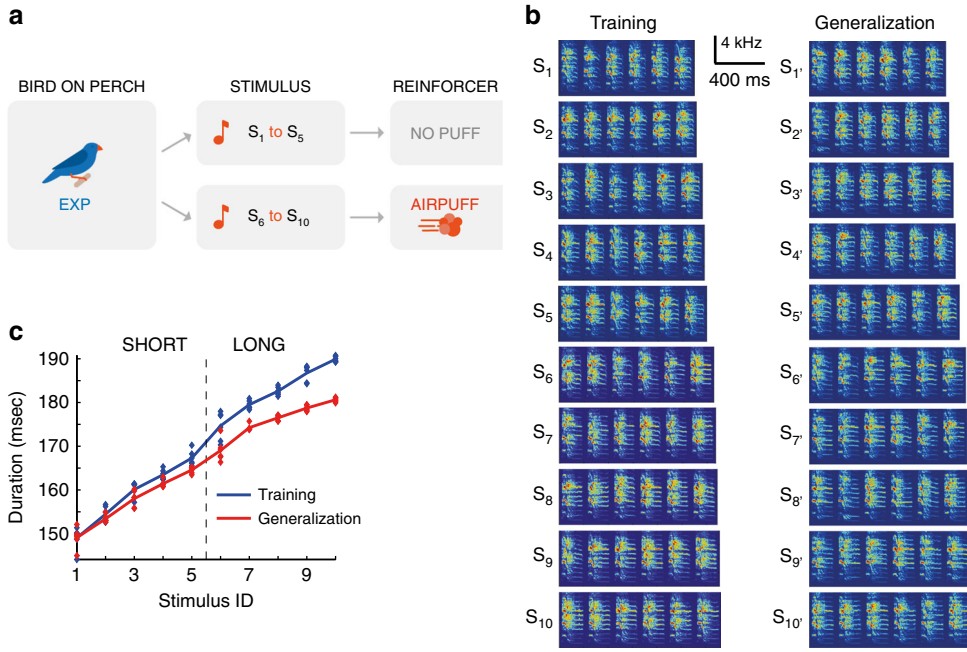

**Fig. 1** A Go-NoGo auditory discrimination task. **a** When the experimenter (EXP) was on the perch continuously for 3.5 s, an acoustic stimulus $S_i$ ($i = 1,..,10$) was randomly chosen and played through a loudspeaker. In this example, the long stimuli ($S_6$ to $S_{10}$) were followed by an air-puff aimed at the experimenter. Experimenters were expected to learn to avoid the air-puffs by escaping the perch in puffed trials, and staying on the perch in unpuffed trials. **b** Log-power spectrograms of all ten stimuli in the training set ($S_1$ to $S_{10}$, left) and in the generalization set ($S'_1$ to $S'_{10}$, right). All stimuli were composed of a string of six renditions of a particular song syllable. **c** Syllable durations for the ten stimuli in the training set (blue) and generalization set (red, dots indicate individual syllable renditions). Either the long stimuli or short stimuli were followed by an air-puff

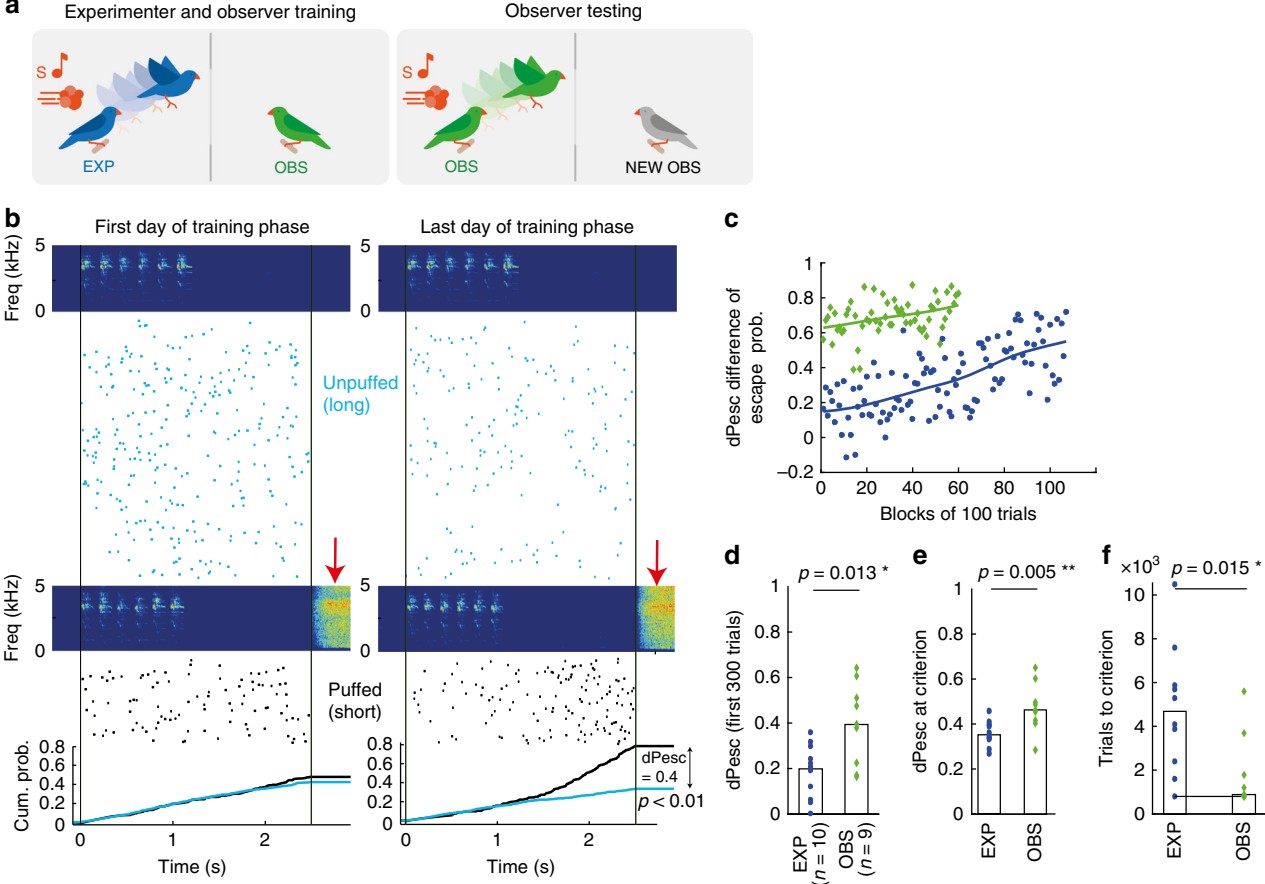

**Fig. 2** Rapid learning in observers. **a** Experimental design: Observers (OBS) were separated from experimenters (EXP) by a screen that restricted visual interactions to the perch equipped with the air-puff delivery mechanism. During a training phase (left panel), OBS watched EXP perform the task. Thereafter, the knowledge learned by OBS was tested in the same paradigm with a new (naive) OBS (right panel). **b** On the first training day (left), this example EXP showed roughly equal densities of escapes (rasters) for unpuffed (light blue) and puffed trials (black). The air-puff sounds are visible in spectrograms of microphone recordings (red arrows). On the last day of training (right), the cumulative escape probabilities (black and blue lines, bottom) discriminate puffed from unpuffed trials (z-test of individual proportions, $p < 0.01$). **c** Difference of escape probabilities (dPesc) during training of an EXP (blue) and during testing of its OBS (green). Solid curves are smoothing spline fits (parameter: $5*10^{-5}$). **d** Bar plot of dPesc, showing that OBS (green diamonds, $n = 9$) discriminate significantly better in the first 3 blocks of their testing than EXP (blue circles, $n = 10$) in the first 3 blocks of their training (EXP ≠ OBS, $p = 0.013$, Wilcoxon ranksum). **e** Upon reaching the learning criterion, the average dPesc (3-block average) in OBS is significantly larger than in EXP (EXP ≠ OBS, $p = 0.005$, Wilcoxon ranksum). **f** OBS reach the learning criterion in fewer trials than EXP (EXP ≠ OBS, $p = 0.015$, Wilcoxon rank sum). The bars indicate the median across birds in each group, the black line indicates the lower bound (800 trials)

attained a statistical performance criterion (based on the perching behavior in the most recent 800 trials, see Methods and Supplementary Figure 1d) after 4700 [800, 10500] trials (median [range], $n = 10$ birds). This criterion defined the end of the training phase, at which time EXP displayed a dPesc of 0.35 [0.27, 0.46] (median [range], dPesc averaged over the last 3 blocks of training, including the criterion block). After the training phase (or observation phase for observers), the experimenter was replaced by the observer (OBS) and a naive bird was placed in the observer's cage. Then we began testing the OBS using the same pre-training and training paradigms it was previously allowed to observe. We refer to the training of observers as testing, Fig. 2a right panel.

At the beginning of the testing phase (first 3 testing blocks), OBS displayed a significantly higher discrimination performance than EXP at the beginning of their training phase (dPesc in first 300 trials: EXP ($n = 10$) 0.2 [0 0.32], OBS ($n = 9$) 0.4 [0.16 0.64]; EXP – OBS, median difference = −0.22, $p = 0.013$, test statistic = 15; two-sided Wilcoxon rank sum test; 95% CI =

[-Inf −0.1]), Fig. 2d. Surprisingly, OBS' initial performance was no worse than that of EXP who had reached the learning criterion (average initial dPesc = 0.40 in $n = 9$ OBS vs average final dPesc = 0.36 in $n = 10$ EXP). OBS reached the performance criterion nearly instantaneously, in only 900 [800, 5600] trials (median [range], $n = 9$ birds), less than a third of the trials required by EXP (two-sided Wilcoxon rank sum test with alternative hypothesis: EXP ≠ OBS, median difference = 3100, $p = 0.015$ (not exact), test statistic = 75; 95% CI not computed because of ties), Fig. 2f. After reaching the criterion, OBS showed a significantly higher discrimination performance than EXP (dPesc at criterion in OBS = 0.46 [0.29, 0.65]; EXP – OBS, median difference= −0.17, $p = 0.005$, test statistic = 12; two-sided Wilcoxon rank sum test; 95% CI = [−0.2 −0.031]), Fig. 2e.

**Observers generalize poorly compared to experimenters**. To compare generalization in experimenters and observers, first, we allowed generalization observers (GENOBS) to watch

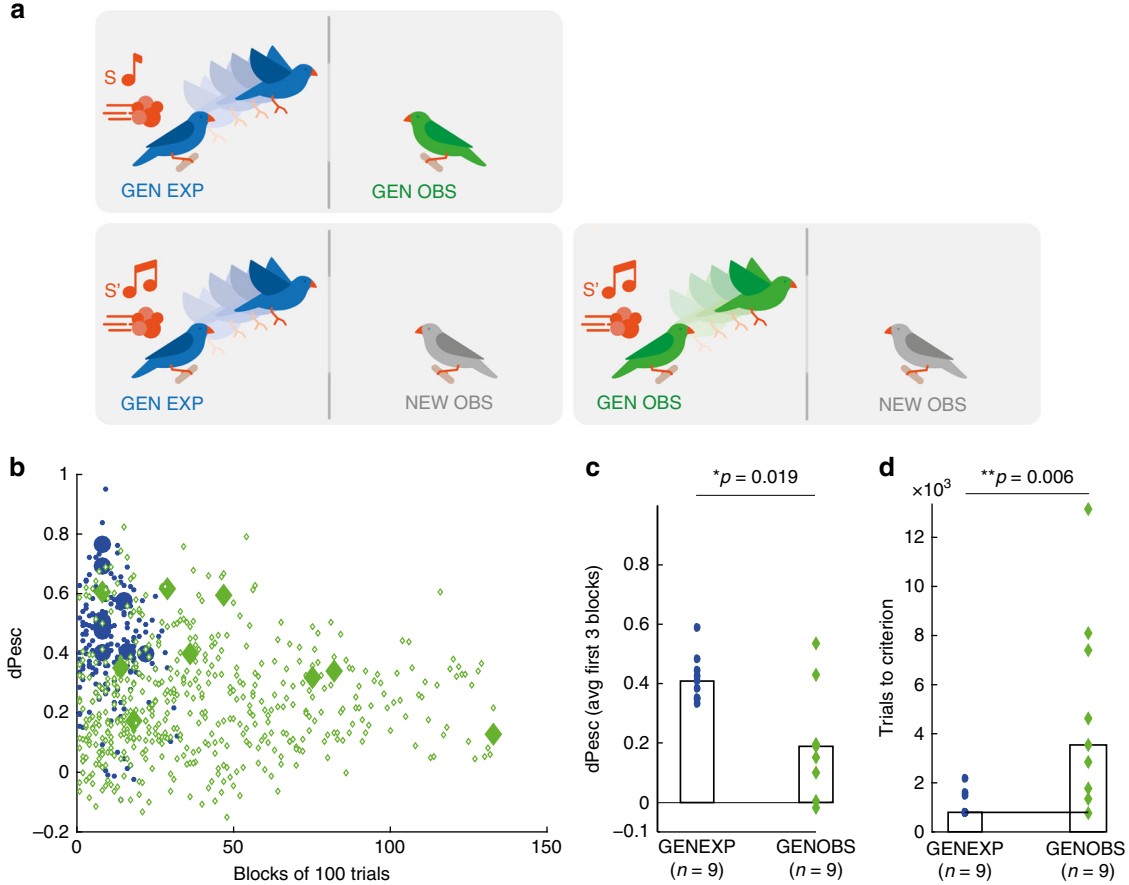

**Fig. 3** Observers are poor generalizers. **a** Generalization experimenters (GENEXP) undergo the same training phase as EXP (training set of stimuli, S, top), after which they are exposed to the generalization set of stimuli S' during the testing phase (bottom left). Generalization observers (GENOBS) first observe the training set of stimuli (S, top) and then are tested on the generalization set (S', bottom right). **b** Scatter plot of dPesc on the generalization set as a function of block number (100 trials per block) in all birds ($n = 9$ GENEXP, blue dots; $n = 9$ GENOBS, green diamonds), the criterion block is represented by larger solid symbol. **c** GENEXP discriminated stimuli in the generalization set better than GENOBS (GENEXP $\neq$ GENOBS; $p = 0.019$, Wilcoxon ranksum). Symbols indicate dPesc averaged across the first 3 blocks of the testing phase, bars represent medians across animals. **d** GENEXP reached the criterion faster than GENOBS (GENEXP $\neq$ GENOBS; $p = 0.006$, Wilcoxon ranksum). Symbols indicate trials to criterion, bars represent medians

generalization experimenters (GENEXP) learn to discriminate the stimuli in the training set, after which we tested both groups of birds on the generalization set of stimuli, Fig. 3a. Contrary to our findings on the training set, GENOBS initially showed significantly poorer discrimination on the generalization set (average dPesc over the first 3 blocks (median [range]) in GENEXP: 0.41 [0.34, 0.6] and in GENOBS: 0.2 [−0.02, 0.54]; GENEXP − GENOBS, median difference = 0.24, $p = 0.019$, test statistic = 67, two-sided Wilcoxon rank sum test, 95% CI = [0.016, 0.36]), Fig. 3b, c. GENOBS also took more time than GENEXP to reach criterion (3600 [800, 13300] trials in $n = 9$ GENOBS versus 800 [800, 2200] trials in $n = 9$ GENEXP; GENEXP—GENOBS median difference = −2800, $p = 0.006$ (not exact), test statistic = 10, two-sided Wilcoxon rank sum test; 95% CI not computed because of ties), Fig. 3d.

GENOBS needed more trials to reach the learning criterion than did OBS (GENOBS—OBS, median difference = 2100 trials, $p = 0.044$, test statistic = 63.5, two-sided Wilcoxon rank sum test), demonstrating that observers reacted to small differences between stimuli from the training and generalization sets. Thus, overall, observers seemed to associate the perch-escape behaviors by experimenters much more exclusively with the presented auditory stimuli than did the experimenters themselves, who associated the air puffs inclusively with the stimuli (to include similar stimuli from the generalization set).

We inspected the escape behaviors of observers and experimenters. We found that after reaching the learning criterion, EXP and OBS displayed similar perch escape strategies. That is, they tended to abruptly increase their perch escape rates just before air-puff onsets (Supplementary Figure 2a, b), suggesting that birds responded by learning to escape the air puffs rather than by learning to stay when no puff was imminent.

**Observers do not learn through passive perceptual processes.** We set out to characterize the requirements for observation learning. To test whether observers learned from experimenters' actions in response to the air-puffs, we allowed experimenter and observer pairs to experience several thousand (mean ± standard deviation = $7.5 \pm 3.6 \times 10^3$) stimulus playbacks including the sound of air-puffs, but not the tactile sensation of the puffs. We realized this perceptual paradigm by directing the air outlet away from the experimenters, Fig. 4a. Consequently, experimenters never experienced the air-puff as a force against their body. We refer to observers in such pairs as perceptual learners (PLs), because they could potentially learn from the pairing of stimuli with air-puff sounds.

Experimenters in this perceptual paradigm never produced dPesc values different from 0 (average dPesc after 5000 training trials in 3 experimenters: [−0.065, −0.002, 0.007], $p = 0.81$,

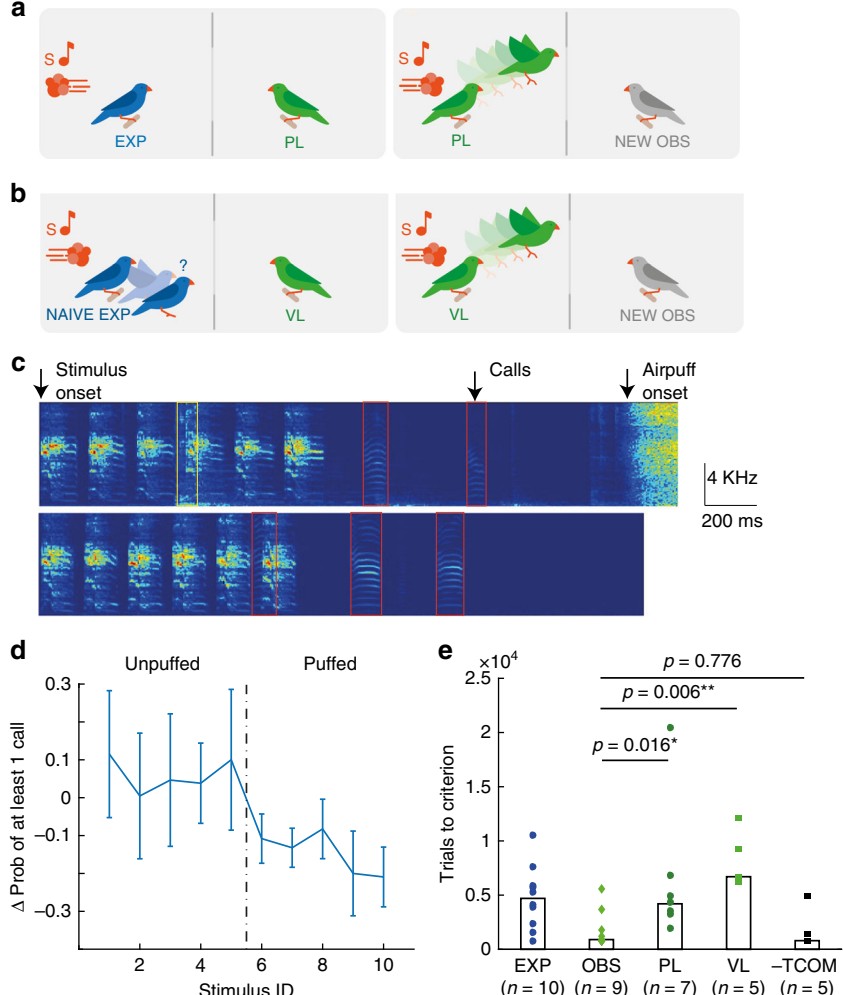

**Fig. 4** Observers learn from behaving experts even in the absence of vocal interactions. **a** Perceptual Learners (PLs, $n = 7$ birds) first observed a naive experimenter trigger several thousand trials in which the air-puff was directed away from the experimenter's body (left). Thereafter, they were tested using air puffs (right). **b** Valence Learners (VLs, $n = 5$ birds) observed experimenters that never reached the criterion (left). Additionally, three VLs were exposed to stimulus-contingent air puffs prior to observation. Thereafter, VLs were tested just like OBS (right). **c** Spectrograms of microphone recordings of puffed (top) and unpuffed (bottom) trials. Vocal exchanges (calls, red rectangles) frequently occurred during the task. Wing flaps were also audible (yellow rectangle). **d** Difference in the probability of observing a call in the delay period and stimulus period (Delay – Stimulus) for the ten stimuli ($n = 6$ observers). Here, S1 – S5 are unpuffed. Error bars show standard deviation around mean. **e** Both PLs (circles) and VLs (squares) required significantly more trials than observers (light green diamonds) to reach criterion during the testing phase (OBS $\neq$ PL, $p = 0.016$; OBS $\neq$ VL, $p = 0.006$). Observers deprived of acoustic communication with experimenters during trial times are as quick as OBS (-TCOM = OBS, $p = 0.776$, Wilcoxon rank sum test). Bars represent medians

$p = 0.25$, $p = 0.64$, respectively; z-test of individual proportions), hence they did not show the discriminative behavior that we suspected would drive learning in observers. When we tested PLs ($n = 7$ birds) with air-puffs directed at them, they needed significantly more trials to reach criterion than OBS (4200 [1800, 20,300] trials in PLs versus 900 [800, 5600] in OBS; OBS – PL median difference = −2400; two-sided Wilcoxon rank sum test of alternative hypothesis PL $\neq$ OBS, $p = 0.016$ (not exact), test statistic = 8.5; 95% CI not computed because of ties), Fig. 4e. PLs were slower than OBS even after removing an outlier bird (trials to criterion = 20300) in the PL group (median difference = − 2389, $p = 0.032$ (not exact), test statistic = 8.5, 95% C.I not computed). PL performance at criterion was comparable to OBS performance (0.33 [0.064, 0.63] in PL versus 0.46 [0.29, 0.65] in OBS; OBS- PL; median difference = 0.16, $p = 0.142$, test statistic = 46, two-sided Wilcoxon rank sum test, 95% CI = [−0.077 0.338]) and was not statistically different from

performance in EXP (EXP – PL; median difference = 0.06, $p = 0.41$, test statistic = 26, 95% C.I = [−0.2 0.2], two-sided Wilcoxon rank sum test). The absence of rapid learning in PLs suggests that learning in OBS required an experimenter engaged in the task and responding to air puffs.

**Observers do not learn from naive experimenters.** We expected observation learning to be most effective when information is provided by an expert. To probe for sensitivity to experimenter performance, we tested a group of Valence Learners (VLs, $n = 5$) that observed naïve experimenters who did not reach the performance criterion within (on average) 5600 [4360, 11436] trials. These naïve experimenters were hit by air puffs on average 539 times out of 1000 puffed trials, and escaped in unpuffed trials on average on 400/1000 trials. In addition, to give VLs direct experience of the reinforcer (its valence), 3/5 of these VL

birds were initially exposed to air puffs (approximately 500 strong 1-s air puffs, see Methods). When tested, VLs were much slower than OBS to reach the learning criterion (trials to criterion in VL [$n = 5$], median [range]: 6700 [6200, 12,100] versus OBS [$n = 9$]:900 [800, 5600]; OBS—VL median difference = −5500, two-sided Wilcoxon rank sum test of alternative hypothesis VL ≠ OBS, p = 0.006 (not exact), test statistic = 0, 95% CI not computed), Fig. 4e. The performance of VLs at criterion was lower than the performance of OBS (dPesc for VL [$n = 5$]: 0.26 [0.13, 0.45] versus for OBS [$n = 9$]:0.46 [0.29, 0.65], median difference = 0.21, p = 0.007, test statistic = 42, two-sided Wilcoxon rank sum test, 95% CI = [0.034 0.355]), and there was a trend of lower performance in VL compared to EXP (VL—EXP, median difference = −0.11, p = 0.07, test statistic = 10, 95% C.I = [−0.21 0.05], two-sided Wilcoxon rank sum test). The poor testing results in VLs suggest that OBS did not learn by predicting the reward value experienced by EXP and by converting this prediction into an optimal action during testing. Instead, VL behavior suggests that OBS focus on experimenters' discriminative actions, which must necessarily contain the information required for observation learning. In combination, PLs and VLs emphasize the importance of experimenters' discriminative actions for observation learning.

**Vocal exchanges are not required for observation learning.** Given the importance of experimenter actions, we speculated that rapid learning in OBS could depend on vocal exchanges between EXP and OBS through calls occurring during and following stimulus presentation, Fig. 4c. Indeed, on the last day of the training phase, when EXP had reached the learning criterion, we found a difference in calling behavior between puffed and unpuffed trials. In six EXP-OBS pairs (on one day each), we inspected calling rates (defined as the probability of observing at least one call) during the stimulus period (from stimulus onset to stimulus offset) and during the delay period (defined from stimulus offset to air-puff onset), Fig. 4d. In puffed trials, the calling rate was lower in the delay period than in the stimulus period: stim call probability 0.44 [0.14, 0.88] vs delay call probability 0.27 [0.07, 0.45], median difference (Delay – Stim) −0.15, $p = 3.8 \ast 10^{-6}$, test statistic = 0, two-sided Wilcoxon sign rank test, $n = 6$ EXP-OBS pairs. In unpuffed trials, there was merely a trend of reduced calling during the delay period: stim call probability 0.35 [0.08, 0.65] vs delay call probability 0.36 [0.16, 0.91], median difference 0.08 (Delay – Stim), $p = 0.052$, test statistic = 327, two-sided Wilcoxon sign rank test, $n = 6$ EXP-OBS pair. In combination, the reduction in calling rate was much more pronounced during puffed trials: difference in median call probabilities for puffed (Delay – Stim) – unpuffed (Delay – Stim) = −0.23, $p = 10^{-6}$, test statistic = 592, two-sided Wilcoxon rank sum test, $n = 6$ EXP-OBS pairs). Hence, the significant reduction in calling rates during puffed trials could signal the imminent arrival of an air puff.

To test whether observers used calls as a learning cue, we housed experimenters and observers ($n = 5$ pairs) in separate soundproof boxes and gave them visual access to each other by virtue of two adjacent windows. Moreover, to trigger social interest, we allowed birds to vocally interact with each other using a custom digital communication system composed of two microphones and loudspeakers and an echo cancellation filter (Supplementary methods). We suppressed vocal exchanges during stimulus presentation by interrupting the communication system from stimulus onset to air-puff offset. We termed the observers in this paradigm no-trial-communication learners (-TCOM). Despite elimination of vocal interactions during the discrimination task, we found that -TCOM acquired stimulus-

discriminative information in amounts comparable to OBS (trials to criterion: -TCOM ($n = 5$): 800 [800, 4900]; OBS ($n = 9$): 900 [800, 5600]; OBS – TCOM median difference = 0.00001, $p = 0.77$, test statistic = 25, two-sided Wilcoxon rank sum test, 95% CI = [−1200 2300]), Fig. 4e. Hence, it follows that OBS did not require immediate vocal interactions. They could learn from visual displays only or from vocal exchanges following trials.

**Regularized logistic regression differentiates OBS from EXP.** Observer behavior was reminiscent of a machine learning system that overfits the training data and generalizes poorly because it contains too many parameters and is trained on too few examples. In this sense, observers seemed to lack regularization, which is an umbrella term for all kinds of processes that prevent overfitting by introducing additional information, for example to use as few nonzero parameters as possible during fitting. Essentially, regularization methods improve generalization performance by dynamically regulating the use of parameters and of training data[16,17].

In the context of our findings, these insights from statistical learning theory suggest that direct experience of the reinforcing learning cue is associated with regularization whereas observation is not. We tested the hypothesis that regularization could set the divide between experimenter and observer behaviors, by training a simple artificial neuron with a logistic activation function to discriminate between the two stimulus sets, Fig. 5a. The neuron received input from a group of at least 22 input neurons tuned to diverse sound features such as amplitude, pitch, duration, and Wiener Entropy, collectively defining the feature set used in Sound Analysis Pro (SAP), a popular birdsong analysis software[18] (Supplementary Figure 4 and Supplementary Table 1). To model observers, we trained the neuron to fire during puffed stimuli and to remain silent during unpuffed stimuli. We used a gradient descent learning rule that maximizes the likelihood of correct discrimination (Methods). We found that the discriminative performance of the 'observer' neuron increased rapidly to the theoretical limit on the training set, but when we interrupted the training at any time and evaluated the neuron's performance on the testing set, we found poor generalization, Fig. 5b. The reason for poor generalization was that the neuron based its classification on exceedingly many sound features that by chance were slightly informative about the reinforcing air-puff, Fig. 5e and Supplementary Figure 4.

We then modeled experimenters by endowing the learning rule with L1 regularization. L1 regularization implements a conjunctive minimization of summed absolute synaptic weights[17] that we implemented at each synaptic weight update as a small reduction of synaptic weights by an amount $\lambda$[19]. We used L1 regularization because it is very simple (subtractive) and because it allowed us to formulate a mechanism that dynamically regulates the regularization parameter $\lambda$ in proportion to reward prediction error (Methods), known to be signaled in the vertebrate brain by a class of dopaminergic neurons[20–22]. According to our proposal, regularization (weight reduction) increases when the bird suddenly receives less reward than expected, as in experimenters that get hit by an air puff for the first time. Our proposed mechanism is such that when experimenters reach a high rate of success, the reward prediction error reaches zero in expectation, which settles the value of $\lambda$, Fig. 5c. The observer brain would not modulate $\lambda$ because observers do not directly experience rewards and punishments during the experimenter training phase.

We found that interrupting the training process of the regularized neuron at any time resulted in roughly equal performances on both training and testing stimulus sets, Fig. 5d,

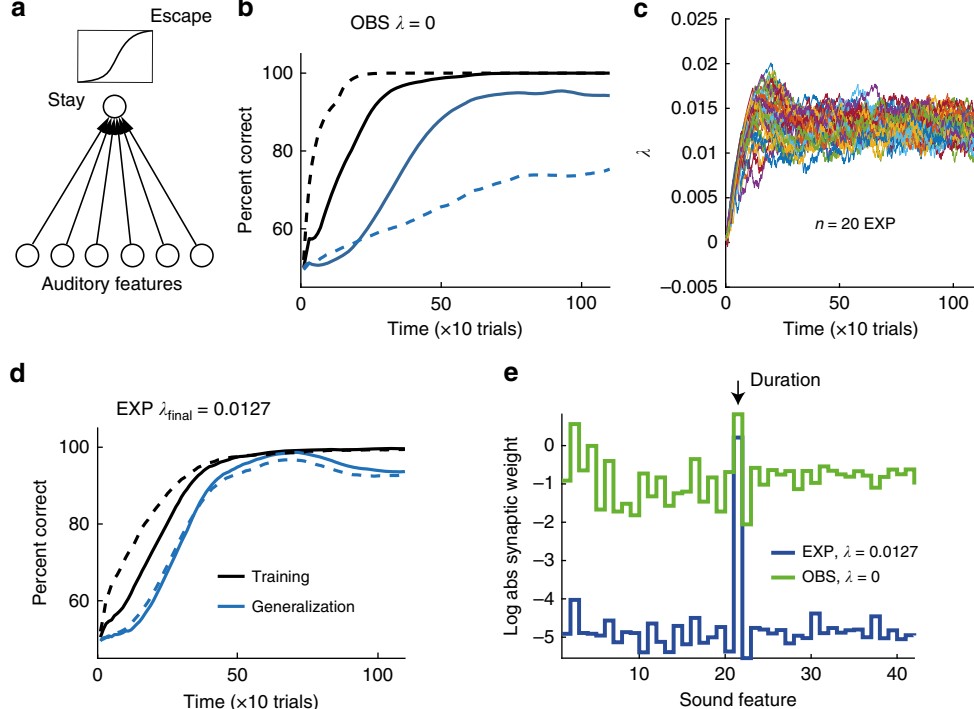

**Fig. 5** Regularization can explain the performance differences between experimenters and observers. **a** The model neuron triggers escapes from the perch based on the logistic response to a set (here 6) of auditory features. **b** When an 'observer' neuron responding to 42 auditory features is modeled without L1 regularization ($\lambda = 0$), the percent correct classification (PCC) on the training set (black line) increases rapidly but the PCC on the generalization set (blue line) increases much more slowly. Adding an extra 100 auditory features of frozen white noise (dashed lines) accentuates the contrast between fast learning and slow generalization. **c** The dynamics of the regularization penalty $\lambda$ under the reward prediction error rule (each color is one simulation run, $n = 20$ simulated birds). **d** When the 'experimenter' neuron is trained with L1 regularization (with dynamic estimation of $\lambda$, the final value of $\lambda$ (on average) $= 0.0127$), the curves reporting PCC on training and generalization sets increase slowly but at roughly matched rates. Increasing the number of frozen noise inputs by 100 has almost no effect on PCC curves (dashed lines). **e** In observer neurons, the log absolute synaptic weights (green) are roughly uniformly distributed. In experimenter neurons, the synaptic weights (blue) are all near zero except the weight corresponding to syllable duration (auditory feature 21, black arrow). Thus, the experimenter neuron turns into a duration detector. Curves show averages across 50 simulation runs. Features 23 onwards were frozen noise features

similar to experimenters' behavior. However, the excellent generalization performance came at a cost: Because of the repeated reductions of synaptic weights in modeled experimenters, their synaptic weights and performance on the training set grew only slowly. The main effect of regularization was to concentrate the final synaptic weights on the duration feature, corresponding with our design of stimulus class, Fig. 5e.

We tested other explanations for the differences between experimenters and observers, such as assuming that observers learned from noisy experimenter actions, but found regularization to be the only mechanism that achieved satisfactory simulations results (Supplementary Figure 5).

## Discussion
We introduced a new comparative approach to observation learning of a discrimination task. We quantified task performance in terms of learning speed and ability to generalize, analogous to studies on observational learning of motor tasks, in which performance is quantified in terms of reaction times and generalization across motor effectors[23,24].

We found that zebra finches can learn to discriminatively respond to auditory stimuli by observing expert performers. Experimenter and observers' behaviors were subject to a tradeoff that depended on whether the learning cue was experienced or observed. We inferred this cue dependence thanks to our experiment design in which the stream of auditory stimuli was

identical for experimenters and observers. Therefore, any differences in their abilities to learn and to generalize must have been entirely due to the learning cue, which was an aversive air-puff for experimenters and an observable action for observers. Our findings suggest that an experienced cue favors robust generalization, whereas an observed cue favors rapid learning.

Part of our findings are in line with social learning theories which suggest that to learn from others is a successful strategy with high payoff under a wide range of conditions[25,26]. However, our findings also suggest a limitation to the ubiquitous success of social learning strategies. Namely, we find that social learning can lack robustness when environmental conditions even slightly change. As in the case of children who perform poorly in exams after neglecting their homework, insights gained through observation seem not to transfer well to new task instances.

Currently, there is no reason to think that all forms of observation learning will be subject to lack of robustness. For example, it is not clear that male zebra finches would exhibit similar behaviors given the known sex differences in social learning[27] also in airpuff paradigms[28]. Furthermore, it is not clear whether our findings will generalize to other reinforcers including reward and strong punishment (e.g. by electric shock). It is even uncertain whether to be observed played a role for experimenters' robust learning. In the light of all these possibilities, our work raises the question as to whether there exist some forms of

observation learning that promote robust transfer to new task instances.

Our work raises many interesting questions on the behavioral and neurobiological mechanisms used by observers to acquire stimulus-discriminative information. Behaviorally, observers could learn through social mechanisms of action imitation, of observational conditioning, and of stimulus enhancement, or a combination of these. Note that the definitions of these mechanisms are not strict enough to allow a discrete categorization of social learning in any one study[29]. Our findings de-emphasize some known social learning mechanisms such as perceptual learning (evidenced by PL learners) and simple stimulus enhancement (evidenced by lack of discriminative behavior during pre-testing, Supplementary Figure 3). Our experiments also de-emphasize vocal communication as a mechanism but reveal the importance of vision (-TCOM learners). Overall, the importance of a demonstrating expert suggests that experimenters signal statistical differences between puffed and unpuffed stimuli via their perching behavior such as their rates of leaving the perch. Possibly, observers focused their attention more on the diverse actions of experimenters and their relationships with the stimuli, which is why observers apparently failed to identify the simplest environmental signal that can explain experimenters' behavior, which in our case was syllable duration.

Similar speed-robustness learning tradeoffs as the one we find exist in rapidly evolving artificial systems, in which high discrimination performance tends to be associated with slow learning as an unwanted side effect[30]. The tradeoff we find between robustness in one learning paradigm and speed in another is most closely paralleled by regularization methods that control inference through synaptic weight subtraction. Excellent generalization of experimenters agrees with strongly regularized classifiers whereas fast learning in observers agrees with weakly regularized classifiers. Our work suggests that the benefits of regularization may be inherent to experimenting but not to observing[31].

It is far from clear how a brain could implement dynamic regularization. Our speculative proposal is that the balance between learning and regularizing is controlled by a neuromodulatory signal. Such signals are ubiquitous in the animal kingdom and are well suited to convey the amount of regularization, given that they respond sensitively to external reinforcements and their prediction errors[32–36]. One possibility is that air-puff reinforcers drive changes in regularization via experimenters' escape actions, which is supported by the representation of action-specific reward values in brain areas innervated by neuromodulatory neurons[37]. This proposal delineates a possible neural system for comparative studies of learning from experience and from observation. It has been shown that reward prediction error and reinforcement learning algorithms in general, may be utilized by humans in order to understand the social value of others' behavior[38,39], to feel vicarious rewards from their success or failure[40] or from their approval[41]. We believe that the computational role of reward prediction error can be extended to that of regularization of learning, mediated by neuromodulator systems such as acetylcholine or dopamine. Furthermore, subtractive weight depression through heterosynaptic competition has been observed in the amygdala[31], which provides biological plausibility to L1 regularisation in the brain. We hypothesize that some form of synaptic depression is seen in zebra finches when they are experimenting, but not when they are observing.

The speculative implications of our simulations are that a prerequisite for the evolution of observation learning was a sufficiently large brain capacity that provided rich sensory

representations and put few constraints on usable neural resources for sensory processing. Evolution might have chosen traits in observers that are complementary to those associated with experimenting, explaining the apparent differences in what these learning strategies extract from the sensory environment.

## Methods

In this section, we summarize our experimental procedures, details are provided in Supplementary Note 1.

**Experimental animals.** We used adult (older than 90 days post hatch, dph) female zebra finches (*Taeniopygia guttata*, $N = 51$ females) raised in our colony. All experiments were licensed by the Veterinary Office of the Kanton of Zurich.

**Experimental setup.** We adapted an operant conditioning paradigm using social reinforcement[14,15]. An experimenter and observer pair were placed adjacent to each other in separate cages. The birds could interact from a restricted window in one corner of the cage, forcing them to sit on their respective perches, Fig. 1a. The experimenter perch had a sensor to detect presence or absence and trigger stimulus playback through a speaker. We used strong air-puffs directed toward the experimenter as an aversive reinforcement agent. The puffs motivated the experimenter to escape from its perch during 'puffed' class trials. Details of housing, perch and nutritional requirements are detailed in the Supplementary Information.

**Stimuli.** We created a set of 10 stimuli from the songs of an adult male zebra finch (o7r14) from our colony, Fig. 1c. We computed syllable durations via thresholding of sound amplitude traces. Each stimulus $S_i$ in this set ($i = 1,2, …,10$) was made of a string of six syllable renditions, wherein each rendition was longer than the six renditions in stimulus $S_{i-1}$, Fig. 1b, c. Based on the ten stimuli we defined two stimulus classes: the class 'short' was formed by stimuli $S_1$ to $S_5$, and the class 'long' was formed by stimuli $S_6$ to $S_{10}$. We counterbalanced the air-puff contingency across birds and found no significant difference in performance between "long-puffed" and "short-puffed" groups, Supplementary Figure 3. We use the terms 'puffed' and 'unpuffed' as class labels, irrespective of whether short or long stimuli were reinforced. We refer to the stimulus set $\{S_1,…, S_{10}\}$ as the training set. To create a generalization set we formed another set of 10 stimuli $\{S'_1,…,S'_{10}\}$ from renditions of the same syllable recorded on the very next day, Fig. 1b, c.

**Bird groups and experimental hypothesis.** We used seven different groups of experimenters and observers, as follows:

1. Experimenters (**EXP**, $n = 10$ birds): These birds were trained to escape from the perch prior to arrival of air-puffs. The birds first underwent a pre-training phase in which they were accustomed to the setup, followed by a training phase (see Procedure in the Supplementary Information). Three out of nine experimenters were also tested on a generalization set of stimuli (Generalization phase) once the training phase was completed, Fig. 2a left panel. Each phase ended when the bird's performance reached a set criterion (see Performance measures and statistical criterion).

2. Observers (**OBS**, $n = 9$ birds): Observers were subjected to three phases: an observation phase in which they observed the entire pre-training and training phases of an experimenter, a pre-testing phase (identical to the experimenter's pre-training phase), and a testing phase (identical to the experimenter's training phase), Fig. 2a right. When the EXP finished training, the OBS was moved to the EXP cage, a naive observer was added to the OBS cage and the EXP was removed.

3. Generalizing Experimenters (**GENEXP**, $n = 9$ birds): these birds were tested on the generalization set of stimuli after they had finished the pre-training and training phases on the training set, Fig. 3a. GENEXP birds were moved to another chamber (a new cage with a naïve observer in the adjacent cage). Note: During the training phase, the experimental group GENEXP is a biological replicate of the EXP group. As expected, there was no difference between EXP and GENEXP in learning time or discrimination accuracy on the training set (Trials to criterion, median [range]: EXP = 4700 [800, 10500], GENEXP = 5900 [800, 13600]], $p = 0.45$, test statistic = 90.5, two-sided Wilcoxon rank sum test; dPesc at criterion: EXP = 0.35 [0.27, 0.46], GENEXP = 0.5 [0.21, 0.46], $p = 0.37$, test statistic = 88.5, two-sided Wilcoxon rank sum test).

4. Generalizing Observers (**GENOBS**, $n = 9$ birds): These birds underwent the same observation and pre-testing phases as OBS. Thereafter, during the testing phase, GENOBS were tested on the full generalization set, Fig. 3a. After reaching the learning criterion, GENEXP were moved to another chamber (a new cage with a naïve observer in the adjacent cage) while GENOBS were transferred to the previous cage of the GENEXP with a naive observer in the GENOBS cage. Therefore, GENOBS and OBS were not

treated differently (however GENEXP were moved to an entirely new chamber unlike EXP).

5. Perceptual Learners (**PLs**, $n = 7$ birds): First, PLs could watch an experimenter trigger several thousand stimuli and air-puffs. However, in their case the air-puffs were directed away from the experimenter (oriented downwards outside the cage, Fig. 4a) so PLs never experienced or saw the effect of an air-puff against a bird prior to entering the pre-testing phase. Following this, PLs then underwent the same pre-testing and testing phases as OBS.

6. Valence Learners (**VLs**, $n = 5$ birds): To test for sensitivity on demonstrator performance, we allowed $n = 5$ VLs prior to their pre-testing to observe naive experimenters (who had not reached the learning criterion). After completion of the pre-testing phase, VLs were subjected to the testing phase, Fig. 4b. Three of these five VLs were given an additional several hundred (approximately 500 on average) stimulus-puff pairings (same protocol as EXP training phase) prior to observing the naive experimenter. The reason for exposing these 3 VLs to air-puffs was to let them learn the aversive nature of the air-puff prior to observing the naive experimenter. However, we limited the number of pre-testing stimulus-puff pairings to about 500. Too many pairings would be counterproductive because VLs would start to learn as do experimenters.

7. No Trial Communication Learners (-**TCOM**, $n = 5$ birds): To test against learning in observers from vocal cues (or the lack thereof, Fig. 4c, d) we separated five observers from their experimenters (pre-training and training phases) into an adjacent, acoustically isolated box. These observers (-TCOM) could view the EXP through a window and communicate vocally through a custom (software controlled) communication channel (see Supplementary Methods) except during trial periods defined from stimulus onset to air-puff offset. During trial periods, the observers could only hear the stimuli and the sounds of air-puffs, but no sounds triggered by the experimenter. After completion of the EXP training phase, -TCOM were subjected to the pre-testing and testing phases as for OBS.

**Performance measures and statistical criterion.** For each bird, we partitioned the trials into non-overlapping bins of 100 trials. In each bin we computed the True Positive Rate ($P_T$) as the probability of escape on puffed trials and the False Positive Rate ($P_F$) as the probability of escape on unpuffed trials. Our single measure of performance in each bin is the difference in escape probabilities $dPesc = P_T - P_F$. Within a bin, to decide whether a bird escaped significantly more on puffed trials than on unpuffed trials, we performed a z-test of independent proportions of the following null hypothesis $H_0$ and alternative hypothesis $H_a$:

$$H_0 : P_T = P_F, \qquad H_a : P_T \neq P_F$$

For the z-test of independent proportions, we computed in each bin the z-test statistics as follows (applying Yates' continuity correction):

$$Z_{stat} = \frac{|P_T - P_F| - \left(\frac{n_T - n_F}{2n_T n_F}\right)}{\sigma_\epsilon}$$

$$\sigma_\epsilon = \sqrt{pq\left(\frac{n_T + n_F}{n_T n_F}\right)}$$

$$p = \frac{(n_T P_T + n_F P_F)}{n_T + n_F}, \quad q = 1 - p$$

where $n_T$ is the number of puffed trials in that bin and $n_F$ is the number of unpuffed trials. The p-value $\Pr[z > z_{stat}]$ was computed with the normcdf function in MATLAB (Mathworks Inc); a bin was "statistically significant" if the p-value in that bin was smaller than 0.01 (two-sided test).

**Criterion and trials to criterion.** We used two statistical measures to analyze learning speed: the first measure was used during the experiment to switch the experimental phase of the birds, the second was used to estimate, on a much finer scale, when a bird achieved high and stable discrimination accuracy.

Our criterion for determining when the pre-training/training phases of an EXP ends (pre-testing/testing for OBS) was the following: we performed the z-test based significance test (with significance at $p < 0.01$) on dPesc over an entire day. If daily dPesc was significantly greater than zero for two consecutive days, we switched the phase. This "coarse" criterion allowed us to check performance in a logistically tractable manner and provided high power to the test because the sample size was large (average daily number of trials for $n = 10$ EXP: 722.14 ± 317.5).

For analyzing the data post-hoc, we used the following criterion: z-tests in 8 consecutive bins (of 100 trials each) and checking for 7/8 bins significant.

This latter criterion was used because we wanted to analyze the data on a finer temporal scale and still make sure that the performance was stable. Accordingly, we computed the fraction of 100 trial bins with significant dPesc in a sliding window of 8 bins, Supplementary Figure 1d. When this fraction crossed 90% (= 0.875), we took the last bin in the window as the bin at which the performance criterion was reached ("criterion bin"). 'Trials to criterion' is then simply the number of all trials performed by the bird up to and including the criterion bin. Our conclusions of fast learning and poor generalization in observers were robust to changes in the definition of the learning criterion: For example, results were unchanged when we changed the criterion from 7/8 significant bins to 4/4 bins, or when we computed the criterion in 200-trial bins instead of 100-trial bins (Supplementary methods, Robustness of statistics).

**Group level statistical tests.** No explicit power analysis was used for this study. Our main experimental groups (EXP vs OBS in the article) have a sample size of $n = 9$ (one extra EXP was included because its observer was not tested). We believed this to be an appropriate sample size considering the statistical test we planned on using (non-parametric Wilcoxon test, which is conservative but has higher power for low sample sizes than a parametric test) and the time it took for preliminary experiments to finish.

To compare birds between two groups, we used Wilcoxon rank sum tests (wilcox.test() in R), either one-sided (when there was a concrete alternative hypothesis, e.g trials to criterion in OBS vs EXP, alternative: EXP > OBS) or two-sided (when there was no concrete alternative hypothesis, e.g. trials to criterion in GENOBS vs GENEXP). We first checked (for all bird groups) whether the trials to criterion were significantly non-Gaussian using the Shapiro Wilk test of normality (shapiro.test in R). Because only the EXP and VL trials to criterion were sufficiently Gaussian, we chose to perform non-parametric Wilcoxon tests instead of t-tests. All group level statistical tests and effect size calculations were performed using the R package (R Studio, https://www.R-project.org/).

**Logistic regression with L1 regularization.** We modeled experimenter and observer behaviors using logistic regression, which is a simple machine learning classifier that learns linear decision boundaries. In this model, the bird's behavior (leave or stay on the perch) is computed from the input to the logistic neuron, formed by 21 syllable features provided by Sound Analysis Pro (SAP)[18], a popular software tool for characterizing birdsong and its development (SAP features include mean Wiener Entropy, mean pitch goodness, mean frequency modulation, pitch variance, etc., where mean and variance are computed across syllable duration). Syllable duration formed the 21st feature. Feature 22 was formed by a vector of 1's, endowing the logistic neuron with a bias term. Features 23 and beyond were formed by frozen noise that was randomly drawn from a Gaussian distribution and held fixed for a given syllable. In combination, the total dimensionality of sound-feature vectors $\mathbf{X}$ was $n = 22 + nr$, where nr is the dimensionality of frozen noise. The auditory input $\mathbf{z}_i$ to the logistic neuron associated with syllable rendition $i$ presented during trial $t$ was the z-transformed feature vector $\mathbf{z}_i = \frac{\mathbf{X}_i - \mathbf{m}_{t-1}}{\sqrt{\mathbf{v}_{t-1}}}$, where $\mathbf{m}_t = (1 - \varepsilon)\mathbf{m}_{t-1} + \varepsilon\langle\mathbf{X}_i\rangle_{i=1...6}$ is the running mean feature vector and $\mathbf{v}_t = (1 - \varepsilon)\mathbf{v}_{t-1} + \varepsilon\langle(\mathbf{X}_i - \mathbf{m}_t)\rangle^2_{i=1...6}$ is the running variance vector. Both the running vectors were updated after each trial (here $\varepsilon$ is a small integration rate constant and $\langle.\rangle$ denotes averaging over the six syllables in a given trial).

The partial output of the logistic neuron in response to syllable $i$ signals the probability $f(\mathbf{z}_i)$ of an imminent air-puff, given by $f(\mathbf{z}_i) = \frac{1}{1 + \exp(-\mathbf{W}\mathbf{z}_i)}$, where $\mathbf{W}$ is the synaptic weight vector that forms a scalar product with the auditory input. The bird decides to leave the perch (or to not return to it) if $\left(\sum_{i=1}^{6} f(\mathbf{z}_i)\right) > 3$ (majority vote).

The probability that all six syllables correctly (and independently) predict arrival ($\mathbf{u} = 1$) or absence ($\mathbf{u} = 0$) of an air puff (under a Binomial model) is given by $P_{correct} = \prod_i f(\mathbf{z}_i)^u (1 - f(\mathbf{z}_i))^{1-u}$. We trained the synaptic weights by maximizing $\log P_{correct}$ using gradient ascent (maximum likelihood), $\Delta\mathbf{W} = \eta\nabla_\mathbf{W}(\log P_{correct})$, where $\eta$ is a small learning rate. Replacing the definition of $f(\mathbf{z}_i)$ into this expression, we find for observers the simple perceptron-like learning rule that enforces after each trial the weight update $\Delta\mathbf{W}_{obs} = \eta\sum_i (u - f(\mathbf{z}_i))\mathbf{z}_i$. In simulations, we randomly picked a stimulus at each trial followed by the synaptic weight change. The only two parameters in this model are the integration rate $\varepsilon = 0.01$ and the learning rate $\eta = 0.007$.

To model experimenters, we used the same weight update as for observers and applied an additional weight subtraction $\mathbf{W}_{exp} \rightarrow \mathbf{W}_{exp} - \text{sign}(\mathbf{W}_{exp})\boldsymbol{\lambda}$ on successful trials (leave if puffed and stay if non-puffed), provided the individual synaptic weight was of sufficient magnitude, $|\mathbf{W}_{exp}| > \boldsymbol{\lambda}$ (to prevent small synaptic weights from changing sign). We dynamically regulated $\lambda$ in the following manner:

$$\lambda_t = \max[0, \lambda_{t-1} + \alpha(r_t - \bar{r}_{t-1})],$$

where the reward signal $r_t$ was given by

$$r_t = \begin{cases} +1 & \text{if the decision was correct in trial } t \\ -1 & \text{otherwise} \end{cases}$$

and where $\bar{r}_t = \gamma \bar{r}_{t-1} + (1 - \gamma)r_t$ is a running average estimate of past rewards obtained by the bird. Decisions are correct when birds leave the perch on puffed trials and stay on the perch on unpuffed trials. We set the learning rate $\alpha = 0.00025$, $\gamma = 0.99$ and the initial value $\lambda_{t=0} = 0.0005$.

For the experimenter, the learning cue $u$ is the occurrence ($u = 1$) or absence of an air-puff ($u = 0$). For the observer, we hypothesized that $u$ could be either a) the air-puff cue ($u = 1$ during air-puffs and $u = 0$ otherwise) to which the observer's attention is drawn through the experimenter's behavior (Fig. 5b, d solid lines), or b) the action of the experimenter ($u = 1$ during escapes and $u = 0$ otherwise). In this latter scenario, the observer is provided a noisy supervisory signal due to false positive and false negative decisions of the experimenter (average EXP false positive rate ~ 30%, average false negative rate ~ 35%, $n = 10$ EXP birds). To test hypothesis b, we simulated an observer neuron that on randomly chosen 30% of learning (Training set) trials was driven by erroneous learning cues (i.e. escapes on unpuffed trials and no-escapes on puffed trials), with and without regularization (Supplementary Figure 5 dashed and dot-dashed curves, respectively).

Note that how to set the degree of L1 penalty defined by the regularization parameter $\lambda$ is a common problem in machine learning. This parameter is most often selected using grid search or random search methods, to localize the value that minimizes a cross-validation or held-out validation set error[42]. More sophisticated techniques, such as estimating a Gaussian process regression model between the hyperparameter (such as $\lambda$) and the validation error have recently been developed . However, all these techniques require an evaluation of the validation error for optimization[43], for which there is currently no support in animals and their brains.

**Data availability**. We have uploaded the data and Matlab scripts for our experiment on the data repository ETH Research Collections: https://www.research-collection.ethz.ch/handle/20.500.11850/238568. https://doi.org/10.3929/ethz-b-000238568

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

## Acknowledgements

We acknowledge support from Heiko Hörster and Aleksander Jovalekic for conducting the experiments. We thank Nadja Baltensweiler at University of Zurich, Information Technology, MELS/SIVIC, for drawing Figures 1a, 2a, 3a, 4a, and 4b. We thank Rodney Douglas, Fatih Yanik, Andreas Nieder, and Klaas-Enno Stephan for help with the manuscript. This work was supported by the Swiss National Science Foundation (grants 31003A_127024 and 31003A_156976) and the European Research Council under the European Community's Seventh Framework Programme (FP7/2007-2013 / ERC Grant AdG 268911) to R.H.R.

## Author contributions

G.N. designed the experiment, carried out the experiment, analyzed the data, simulated the model neuron, and wrote the manuscript. J.R. and J.H contributed to the experiment by developing software. R.H designed the experiment, simulated the model neuron, and wrote the manuscript.

## Additional information

**Competing interests:** The authors declare no competing interests.

