## [Peer Review File · Nature Communications]

Reviewers' comments:

Reviewer #1 (Remarks to the Author):

How does knowledge acquired by direct trial and error learning differ from that gained by indirect observation of another individual performing a stimulus-response task? In this study, the authors use zebra finches in an auditory discrimination-aversive conditioning paradigm to compare the (a) speed of acquisition and the (b) generalizability of knowledge learned by direct trial and error versus observation. An 'experimenter' finch and an 'observer' finch are placed in adjacent cages and can interact with each other by sitting on a perch next to a small window. This social reinforcement apparatus has previously been shown to motivate finches to perch next to the window. Using aversive air-puffs, the authors trained the experimenter finch to discriminate between short and long renditions of a song syllable sequence, while the other finch 'observed'. After training, the observer finch was tested on the discrimination task with a new naïve finch taking the place of the observer. The authors found that observers were able to learn the discrimination task and avoid the air-puffs sooner than it had taken the experimenter finches, demonstrating observation learning. However, when tested on a different syllable set, observers were slower than experimenters in generalizing their knowledge to a slightly different sensory stimulus. The authors suggest that the difference in generalizability arises because observers 'overfit' their responses to the initial stimulus set while experimenters do not. Applying ideas from machine learning, the authors show that the rates of learning and generalization seen in experimenters and observers can be captured by a model neuron with and without overfitting using the L1 norm regularization (where one puts a penalty on the sum of the absolute values of synaptic weights). The authors speculate on the plausibility of a neuromodulator-mediated generalization of reward-prediction error to implement regularization in the brain.

This is an interesting study that cleverly uses a social reinforcement paradigm in zebra finches to address the issue of generalizability of direct trial and error learning and observation. It also brings ideas from machine learning into thinking about social learning and proposes a specific mechanism (L1 norm regularization) by which experimenters are able to generalize while observers are not. However, there are a few concerns as outlined below.

Major points:

1. In lines 62-64 of the manuscript the authors say, "We refer to this form of learning as observation learning (which is not meant to imply that observers learn by imitating the actions of experimenters, which is commonly known as 'observational learning')." This is an important point and it would enhance the interpretation of the study to know if observers are indeed imitating the experimenters. Have the authors measured the rates of observers leaving their perch in response to experimenters leaving theirs? Closely housed zebra finches are often influenced by the behavior of neighboring finches making it plausible that observers are more likely to leave their perches in response to experimenters leaving theirs. In any case, measuring observer perch departure rates should not be too difficult and will be informative in two ways. First, it will inform us if observers are imitating, i.e. performing the specific motor act of flying away (there could be differences in learning with and without imitation). Second, if observers indeed fly away in response to experimenters doing so, this might be pertinent in the interpretation of the results that observers do not learn through passively listening to the sound of the air-puff, need expert experimenter models, and do not require vocal exchanges.

2. Past studies report robust generalization in observation learning. Putting the present study in the context of the previous literature will benefit the reader. It will also help clarify the limits on the generality of the present study's finding that observation learning does not generalize. Specifically, in

lines 371-373, the authors say, "But our work raises the question as to whether there exist some forms of observation learning that promote robust transfer to new task instances." The question of the generalizability of knowledge learned by trial and error and observation has been addressed previously both in humans (e.g. Cameron et al. 2015, Buchanan and Wright, 2011, Bird and Heys, 2005, Broeren et al. 2011 and references contained therein) and in animals (e.g. Mineka et al. 1984), but this work is not discussed or cited.

3. It was not clear if generalization automatically implies regularization or if regularization is one of many ways in which generalization could be achieved. In other words, are the authors hypothesizing that generalization is achieved through regularization in the brain? If so, what are the alternatives? Similarly, was L1 norm regularization chosen due to its simplicity or is there something that L1 norm regularization captures that other kinds of regularization do not? It will benefit the reader if these issues are briefly discussed.

Minor points (including typos and comments on clarity):

1. Line 46: 'hanks' should be 'thanks'.
2. In lines 71-72, the authors say, "The long stimuli S6 to S10 were followed by an air-puff aimed at the experimenter." Then in lines 76-77, they say, "Either the long stimuli or short stimuli were followed by an air-puff." The first statement is confusing.
3. Line 104 (fig. 2C): What is the purpose of the smoothing spline fits with parameter 0.3? Are these oscillations significant and should we take them seriously? If not, the solid curves are distracting.
4. Line 106 (fig. 2D): Many of the larger solid symbols are lost in the smaller symbols and are barely visible in the plot. Consider changing the plotting scheme for clarity. See also fig. 3B.
5. Line 111 (fig. 2E-G): What do the whiskers indicate? See also fig. 3C-D.
6. Line 251: Fig. 4G should be Fig. 4E?
7. Line 310: "the synaptic weights (black)..." should be ...(blue)?
8. Lines 322-323: "...in proportional" should be "...in proportion" or "...proportional"?
9. Line 337: What is alpha?
10. Lines 487-489: Is it implied that the number of stimulus-puff pairings given to the 3 VL birds is too small for them to learn to escape the puff? This needs to be made clear and explicit.
11. Lines 429-430 in main paper and lines 5-6 in sup.: There is a discrepancy in the number of animals used for the study. It says 46 in the main paper and 51 in the supplement.
12. Supp. Line 6 and main paper lines 62-64: Whereas in the main paper the authors stress that they only claim to be studying 'observation' learning and not 'observational' learning, they say they are studying 'observational' learning in the supplement. This needs to be consistent.
13. Supp. Line 15: Is 'scuttle bone' meant to be cuttlebone?
14. Supp. Line 77: 'of' should be 'to'?

15: Supp fig 1D: If the y-axis is 'fraction of 8 blocks significant', does it make sense to have data points for the first 7 blocks?

16. Supp. Fig. 3B-C: It was not clear why the EXP and OBS are leaving the perch earlier during the un-puffed trials? Please clarify.

References:

Cameron, G., Schlund, M. W. and Dymond, S. (2015) Generalization of socially transmitted and instructed avoidance. *Front. Behav. Neurosci.* 9, 159.

Buchanan, J. J. and Wright, D. L. (2011) Generalization of action knowledge following observational learning. *Acta Psychologica* 136, 167-178.

Bird, G. and Heyes, C. (2005) Effector-dependent learning by observation of a finger movement sequence. *Journal of Experimental Psychology: Human Perception and Performance* 31, 262-275

Broeren, S. Lester, K. J., Muris, P. and Field, A. P. (2011) They are afraid of the animal, so therefore I am too: Influence of peer modeling on fear beliefs and approach avoidance behaviors towards animals in typically developing children. *Behaviour Research and Therapy* 49, 50-57.

Mineka, S., Davidson, M., Cook, M. and Keir, R. (1984) Observational conditioning of snake fear in rhesus monkeys. *Journal of Abnormal Psychology* 93, 355-372.

Reviewer #2 (Remarks to the Author):

Comments to "Learning to perform auditory discrimination from observation is efficient but less robust than learning from experience" NCMMS-17-27463

This is an interesting manuscript that shows something that would intuitively be unexpected, that observational learning can be faster than learning from own experience. As the learning mechanism is very basic (conditioning) and the model species is not "too cognitively advanced" (zebra finches) I suspect that this could depend on some basic mechanism. It is also interesting that this quick social learning allows for less generalization compared to learning from own experience. I think that these are important and novel results; however, I also have some problems with the manuscript. One is the volume and detail of the analyses. Many ideas and side-tracks are tested and controlled for, it takes two days to read this manuscript, and to go through all the analyses and ideas in detail would take a week. As I cannot spend that much time on this, I will keep my comments on a general level. If the authors concentrated more on the main findings, I think this would increase readability a lot.

Also, I lack a discussion of why observational learning should be quick but with poor generalization. Is this only a learning trade-off? Is this something typical for zebra finches? Something general? For small finches, couldn't it be important to react quickly to what others are doing? Imagine an airborne predator attacking zebra finches foraging on the ground. A quick response to what others are doing could then be decisive for survival.

The discussion of machine learning is very detailed, and it feels somewhat speculative. Maybe you could reduce it?

Detailed comments:

L 42: what is "sensory learning"? What would "non-sensory learning" be?

L 44: is "cue dependence" the right word? Sounds like you have studied different types of similar

cues, for example different types of sound cues.

L. 46: You are probably correct in that zebra finches are useful models, but their ability to detect subtle sound differences is probably not better than other passerines. Maybe you should add that they are easy to breed in captivity.

L. 50: Don't you think that they already could discriminate the sounds and that you trained them how to react to them?

L. 51: Zebra finch sounds are probably the most appropriate ones for a study such as this. Do you think that there could be any "bias" in them, for example that the finches could be genetically pre-conditioned to react to them in some way?

L. 55: Did the experimenter finches ever give any alarm or aversive calls at the puff that could work as non-controlled cues?

L. 68: These measures are also important outside the machine learning community!

Figure 1. The most vital part of the figure is C) do you really need the rest? It is also explained in the text.

L. 71-72 and 76-77: Why did you consistently use the long sound in association with the air puff and either the long or short in c)?

L. 81: Why is this pre-training rather than different types of training?

Fig 2. This figure has very much information in it. Is everything necessary, for example a) and b)?

L. 96: Why were they separated by a screen? Could the observers see the reaction of the experimenter at the puff?

L. 119: You mean one-sided? How can you use a directional hypothesis for something that was not beforehand expected? Even if hypotheses are directional, and the opposite outcome doesn't make sense, most journals want two-sided tests. I would have expected that own experience would give faster learning. Also, overall, even if you calculate effect sizes and make power analyses, sample sizes are really small.

L. 132: How can they be exposed to a pre-training phase when they have already been trained to be observers? Why was this pre-training necessary?

L. 136-140. Is the test statistic T from Wilcoxon? You have a dispersion measure at the mean but also give 95% CI separately. Why? Are they different?

L. 141: Is this one- or two-sided? Why don't you state this here as you do above?

L. 161: What do you mean by similar here? How did you compare them? Video analyses?

L. 165: What would a passive perceptual process be here?

L. 169. Several thousand? Don't you re-condition their response by this?

L. 178-181: How can you calculate a dispersion measure but not 95% confidence intervals?

L. 191: What does this sentence mean? "Observers learn from behaving, expert experimenters,..." It does not make sense to me.

L. 192-193: Could the birds themselves trigger this?

L. 204: "Expert" models? This sounds like there were some really good ones among the ones that learnt. Would the ones that did not learn induce any learning at all? I guess they did if they were closer to EXP than OBS?

L. 229-230: Rather than "speculated", did you not actually want to control that it did not?

L. 229-233: What type of calling? Alarm calls?

L. 239: Why do you use a parametric test here, but otherwise non-parametric? Was the t-test two-sided?

L. 245-246. This is a nice design!

L. 254: Would anyone claim this? Do you really need to test for it?

L. 287: I would suggest that both own experience and observation gives experience.

L. 271-352: I would be cautious with making too strong inferences from machine learning. I think the authors have a point here, and that comparisons can give us insights. But machine learning is just machine learning and we don't know to what degree it actually resembles what goes on in real bird brains. When machine learning is used to model learning in animals it will in most cases be a

simulation of learning, not exactly the same process. I am not at all sure that the machine learning dilemma (too specific but rapid learning vs slower but more useful general learning) should be directly applicable to what we see in the zebra finches. It could be something like that, but it could also be something completely different. In what way are zebra finches evolved to react on social learning cues? By the way, do you really need this detailed machine-learning section?

L. 305-306. Is this because more noise makes generalization harder?

L. 319-321: This does not say anything to me. Could you explain it in a simpler way? The same is true for L. 345-347.

L. 355: Could they not discriminate these stimuli already before the experiment? Do you not mean "learn to react to auditory stimuli..."

L. 356: have you actually demonstrated this trade-off? Your results suggest it, but that is not the same thing. What was the trade-off, learning speed vs generalization ability?

L. 361-362: Why would this be so? Is it only a trade-off depending on the learning type or could the birds have natural, evolved responses that differ depending on ecological factors?

L. 366-367. Yes, in this system. If you look at a more cognitively advanced bird, such as a raven, social learning is probably different, occurring on a high cognitive level with lots of generalization (like you say on lines 370-371).

L.395: How did the amygdala come into this? As I know it, it is a site for emotional memory.

L. 405-406. Well, maybe you should not discuss it in such detail then.

L. 410: Another possibility could be that it has nothing to do with regularisation at all.

L. 470: What is a biological replicate?

L. 539: I would say, in case of negative results, power tests are always useful. Also, $n = 9$ is still a very small sample size. Even if you use a non-parametric test a larger sample size would have been good. It is hard to motivate a sample size smaller than 10 with statistical reasons.

Reviewer #3 (Remarks to the Author):

This study aims to study the mechanisms of observational learning in birds (zebra finches). The ms is composed of 2 parts: an experimental part and a modeling part.

The authors used an operant conditioning task with mild punishment (air puff). During the test phase, the birds were trained to discriminate sound stimuli ($n = 10$) classified into 2 categories. These sound stimuli were constructed from multiple copies of a song syllable of a male finch. The authors selected 10 copies of different durations and prepared each stimulus by repeating the syllable 6 times with constant inter-syllabic intervals. They created 2 categories of stimuli: short and long. During a test phase, the bird had to learn to discriminate the sounds of the 2 categories. During the generalization phase, the bird had to learn to discriminate new stimuli. These new stimuli were constructed in the same way as those generated for the test phase, but using exemplars of the song syllable of the same male recorded the next day.

In a first experiment, a female observer (OBS) observed the tested female (EXP) during the learning phase, before being tested in the presence of a new female naive observer. The authors observed that OBS females learned the task of auditory discrimination faster than EXP females.

In a second experiment, a female observer (GENOBS) observed the tested female (GENEXP) during the learning phase. The two females were then placed each with a new female naive observer. In this situation, the authors observed that GENOBS females took longer to learn to discriminate the set of stimuli used for generalization. The authors interpret this result by the fact that observers (OBS) react to small acoustic differences between the stimuli used for training and generalization whether EXP birds focused more on the duration of the stimuli.

In a third experiment, the authors used the same paradigm as in the first experiment but they directed the airflow so that it did not affect the tested female (EXP) while another female watched her (PL). Thus, only the sound of the air puff was audible during the learning phase. Not surprisingly, the

bird EXP did not learn to discriminate the stimuli of the 2 categories. When the PL birds were tested with the airflow directed at them, they took longer than the OBS birds to discriminate the stimuli broadcast during the training phase. This difference suggests that OBS learned the discrimination task by observing the behavioral response of the EXPs during the air puff, and not by being passively exposed to the noise of the air puff.

After observing rapid learning in OBS and slow learning in PL, the authors questioned whether experimenters should be experts to induce learning in observers. They exposed 5 females (VL) to experimenters who had not yet achieved the task of discrimination. They also exposed 3 VL to air puffs (about 500) following the same protocol used for the first experiment, before giving them the opportunity to observe experimenters who had not yet succeeded in the task of discrimination. The authors observed that the performance of the VL birds was weak suggesting that the OBS did not learn by predicting the value of the reward and by converting this prediction into optimal action. On the contrary, the behavior of the VLs indicates that the OBS focused their attention on discriminative actions of experimenters.

In another experiment, the authors investigated the influence of vocal communication on learning. They observed a change in vocal activity at the end of the training phase between puffed and unpuffed trials. Thus, vocal activity might facilitate learning. The authors used the same procedure as in experiment#1 but they placed the experimenter and the observer in two separate cages. This setup permitted to control the vocal exchanges that were broadcast by a loudspeaker in each cage. The authors prevented the vocal exchanges between the beginning of the broadcast of the sound stimulus and the noise of the air puff. The tested birds (-T-COM) exhibited learning performances comparable to those of OBS of the 1st experiment suggesting that learning was not facilitated by the vocal interactions during the broadcast of the sound stimuli.

In a final experiment, the authors checked whether observational learning was not a simple form of stimulus enhancement. To do this, they modulated the power of the air puff during the pre-testing trials and observed that a weak air puff linked to a sound stimulus did not induce escape behavior in OBS. This means that in addition to the knowledge of the stimuli and the action of the experimenters during the broadcast of these stimuli, the observers also need to make the aversive experience with the air puffs to express their learned knowledge.

In a second part, the authors relied on knowledge in machine learning and statistical learning to propose a simple mechanism that could explain these experimental results. This mechanism is based on a tradeoff between speed of learning and generalization. They used an artificial neuron that received input from a group of at least 22 neurons tuned to various acoustic parameters. This simulation showed that the weak generalization observed in OBS could be explained by taking into account too many acoustic characteristics of the stimuli broadcast during training. Thus, in the neurons observers, the weight of the various acoustic parameters in the classification is balanced whereas for the experimental neurons, the classification is based on the duration of the stimulus which is the criterion for discrimination between the two categories.

Then, the authors manipulated another parameter (regularization parameter) to integrate into their model another aspect linked to the activation of reward circuits during learning (dopaminergic neurons). The idea is that the observers' brain would not modulate this parameter because observers do not directly experience a punishment during the experimenter training phase. From these simulations, the authors conclude that this regularization parameter should be taken into account to obtain a good agreement with the experimental results, and in particular in the observed differences between the performance of observers and experimenters. Using their model, the authors did not succeed in distinguishing between the two possible learning strategies of the observers based on: 1) air puff's auditory cue (the actions of the experimenter directed to the air puff) or 2) the escape behavior of the experimenter.

In the discussion, the authors conclude that an experienced cue favors robust generalization. They also discuss the results of their study in relation to the conceptual and theoretical framework of social learning in animals. They also offer explanations and future directions for exploring the behavioral and

neurobiological substrates of the results obtained.

This is a very elegant study, carefully dissecting the behavioral mechanisms at stake during observational learning in animals. The modeling part nicely complements the experimental part. The statistical treatment seems correct and the data justify the conclusion. This approach is original and the results of this study will be of interest for researchers interested in mechanisms of learning from different fields such as animal behavior, computational neuroscience and evolutionary robotics.

My only concerns are:

1) The use of a single sex (female) in a species for which it has been demonstrated that males and females do not deal with social information in the same way. See for example the study by Katz and Lachlan (Animal Cognition, 2003) showing that social learning of food types in zebra finches is directed by demonstrator sex.

2) The experimental procedure involves a mild punishment (air puff); not a reward or a stronger punishment (electric shock). The authors should clarify whether their model takes into account those aspects that may affect learning.

Here are some more specific comments:

Line 91-92. Was the observer systematically moved to the experimental chamber of the experimenter she observed (same location)? Or is it possible that an observer stayed in her part of the cage (observed chamber) or was moved to another cage of another setup? This change of location seems to be the case for the GENEXP/GENOBS (lines 130-131) and could affect the learning. Please comment.

Line 178. The authors could also compare the results with EXP. They provide this comparison in line 221 but it could be mentioned here.

Line 210. It is not clear whether 3 additional VLs were initially exposed to air puffs or if these 3 VLs are part of the group of the 5VLs. Please comment. See also line 135 of the supplementary information.

Line 232. How could the authors distinguish the vocal activity of OBS from EXP? Was calling rate inspected in OBS (as mentioned in the text) or in EXP? I guess there is a mistake in the text (mention of OBS but not EXP). Please comment.

Lines 327-328. The authors mention the possibility that birds could experience a reward but using this experimental procedure, it is a mild punishment. Please comment.

Lines 381-382. To de-emphasize the importance of vocal communication, the authors should have checked whether auditory cues alone (vocal but not visual communication allowed) could trigger training in OBS. There are also evidence that stress can be coded in animal vocalizations including calls in finches (see Perez et al. Horm. Behav. 2016). There are possibilities that calls produced following a stimulus linked to an air puff might bear information about stress or discomfort.

Line 412. Please correct: neuromodulatory neurons.

Supplementary information, line 7. Male zebra finches do not defend a territory both in the field and in the artificial conditions of the laboratory. This statement is not correct. Male finches produce songs in the courtship context and probably also to facilitate group cohesion. Also, male is the predominantly used sex for neurobiological studies involving song learning since females do not sing. But both male and female finches are used in labs investigating other aspects of the social life.

Supplementary information, line 51. The authors should provide measures of acoustic parameters of these different renditions of the same song syllable that was used for these experiments. They should also provide a short review of the literature regarding the capacities of auditory discrimination in both time and frequency domains in finches. This would help the reader to estimate whether finches are able to discriminate two sounds with a high acoustic similarity such as two different renditions of a same song syllable produced by a single individual.

Discussion. The authors should also discuss about the possibilities that being observed might affect the speed of learning. They took care about it by always having two birds in the experimental setup but there is a possibility that such eavesdropping might affect the bird's behavior. Please comment.

Methods, escape behavior. The authors should provide data about the escape time. Did the

trained/tested bird systematically wait for the end of the stimulus broadcast to escape in case of punishment?

We thanks all reviewers for their very helpful comments that allowed us to improve our manuscript.

Reviewers' comments:

Reviewer #1 (Remarks to the Author):

How does knowledge acquired by direct trial and error learning differ from that gained by indirect observation of another individual performing a stimulus-response task? In this study, the authors use zebra finches in an auditory discrimination-aversive conditioning paradigm to compare the (a) speed of acquisition and the (b) generalizability of knowledge learned by direct trial and error versus observation. An 'experimenter' finch and an 'observer' finch are placed in adjacent cages and can interact with each other by sitting on a perch next to a small window. This social reinforcement apparatus has previously been shown to motivate finches to perch next to the window. Using aversive air-puffs, the authors trained the experimenter finch to discriminate between short and long renditions of a song syllable sequence, while the other finch 'observed'. After training, the observer finch was tested on the discrimination task with a new naïve finch taking the place of the observer. The authors found that observers were able to learn the discrimination task and avoid the air-puffs sooner than it had taken the experimenter finches, demonstrating observation learning. However, when tested on a different syllable set, observers were slower than experimenters in generalizing their knowledge to a slightly different sensory stimulus. The authors suggest that the difference in generalizability arises because observers 'overfit' their responses to the initial stimulus set while experimenters do not. Applying ideas from machine learning, the authors show that the rates of learning and generalization seen in experimenters and observers can be captured by a model neuron with and without overfitting using the L1 norm regularization (where one puts a penalty on the sum of the absolute values of synaptic weights). The authors speculate on the plausibility of a neuromodulator-mediated generalization of reward-prediction error to implement regularization in the brain.

This is an interesting study that cleverly uses a social reinforcement paradigm in zebra finches to address the issue of generalizability of direct trial and error learning and observation. It also brings ideas from machine learning into thinking about social learning and proposes a specific mechanism (L1 norm regularization) by which experimenters are able to generalize while observers are not. However, there are a few concerns as outlined below.

Major points:

1. In lines 62-64 of the manuscript the authors say, "We refer to this form of learning as observation learning (which is not meant to imply that observers learn by imitating the actions

of experimenters, which is commonly known as ‘observational learning’).” This is an important point and it would enhance the interpretation of the study to know if observers are indeed imitating the experimenters. Have the authors measured the rates of observers leaving their perch in response to experimenters leaving theirs? Closely housed zebra finches are often influenced by the behavior of neighboring finches making it plausible that observers are more likely to leave their perches in response to experimenters leaving theirs.

Unfortunately, we measured observers’ perching behaviors only in 3 birds. However, in these 3 birds we did not find correlated escaping between experimenters and observers. First, we computed Pearson correlation coefficient between the binary response vectors (escape/no escape) per trial of a each EXP and OBS pair for the entire training phase of the EXP (Bird pair 1: $r = 0.018$, $p = 0.09$; pair 2: $r = 0.12$, $p = 10^{-52}$; pair 3: $r = -0.05$, $p = 0.0000$; p-values are small due to the large sample sizes ($n = 9075$, 15893 , 6226 trials resp.)). Second, we looked at Pearson correlation coefficients between the binary response vectors only at the end of the training phase (last 1000 trials) in order to compare the performance of the expert EXP bird to its OBS (Bird pair 1: $r = -0.01$, $p = 0.005$; Bird pair 2: $r = -0.024$, $p = 0.44$; Bird pair 3: $r = 0.024$, $p = 0.45$). Therefore, we find no direct evidence that observational learning in the sense of action imitation plays an essential role in our social learning task. We have added a few sentences in the Supplement (*Observer perching behavior*, line 207 – 217, page 10) to this effect.

In any case, measuring observer perch departure rates should not be too difficult and will be informative in two ways. First, it will inform us if observers are imitating, i.e. performing the specific motor act of flying away (there could be differences in learning with and without imitation). Second, if observers indeed fly away in response to experimenters doing so, this might be pertinent in the interpretation of the results that observers do not learn through passively listening to the sound of the air-puff, need expert experimenter models, and do not require vocal exchanges.

We agree in essence with the reviewer. However, note that strictly speaking, lack of correlated escaping in EXP and OBS does not imply absence of an imitation strategy, simply because OBS might not express the imitated behavior at every single trial. This could be similar to song learning, in which young songbirds do not sing along with the tutor at every song rendition, but nevertheless end up imitating the tutor’s song as adults. (This is a continuation of the previous question, the changes to the Supplementary are noted above).

2. Past studies report robust generalization in observation learning. Putting the present study in the context of the previous literature will benefit the reader. It will also help clarify the limits on the generality of the present study’s finding that observation learning does not generalize. Specifically, in lines 371-373, the authors say, “But our work raises the question as to whether there exist some forms of observation learning that promote robust transfer to new task instances.” The question of the generalizability of knowledge learned by trial and error and observation has been addressed previously both in humans (e.g. Cameron et al. 2015,

Buchanan and Wright, 2011, Bird and Heyes, 2005, Broeren et al. 2011 and references contained therein) and in animals (e.g. Mineka et al. 1984), but this work is not discussed or cited.

Here, we provide a summary of the papers suggested by the reviewer. We find that the papers are relevant enough to be cited (since they are concerned with generalization in observational learning), but the tasks and task domains are quite different for a fair comparison with our work.

Cameron et al. 2015 study in humans the generalization of avoidance behavior to novel (but similar) stimuli after fear conditioning to a pair of stimuli. They show that there is a smooth gradient of avoidance response as the stimuli (gray circle of a certain radius, with the radius altered) change from the CS- (unconditioned) to the CS+ (shock conditioned). The gradients are identical in observers and instructed (directly conditioned) learners. This result would speak against our results. However, there are two key differences between our work and Cameron et al. 2015: firstly, the stimuli used are artificial and vary along only one dimension (radius). Therefore, there is no opportunity for observers to “overfit” their learned behavior. Secondly, prior to observation, observers were explicitly informed of the nature of the experiment, i.e., that they would observe another participant perform a behavior in order to avoid a shock. This form of prior knowledge was entirely absent in our experiment.

Bird and Heyes (2005) measured reaction times in observers and controls learning a sequential task in which each participant had to perform a sequence of key presses in response to the sequence of locations where stimuli were presented. They found similar reaction times in both groups when the key sequence used during a test phase was the same as during training. However, observers were significantly faster than controls when the key sequence was changed to require thumb presses instead of finger presses. This was construed as a lack of learning in observers because learning should induce slow adaptation to alterations in required behavior, termed “negative transfer”. The authors concluded that learning in observers was “effector dependent”, i.e. tied to the same effectors used by models who performed the task. Buchanan and Wright (2011) challenged this result in their study by using a relative-phase learning task, where participants learn coordinate elbow and wrist joints. They report successful generalization of relative phase knowledge in observers. Even though these studies report results on generalization in observers, the experiments are not concerned with discrimination of rich, high dimensional natural stimuli and are about observational motor learning, which is not in the scope of our experiments. However, these data do indeed show that generalization through observational learning can be task, modality, and domain dependent.

We cite Bird and Heyes (2005) as well as Buchanan and Wright (2011) in the Discussion section (lines 364-366, page 20) under the context of generalization in observational learning in the motor domain.

3. It was not clear if generalization automatically implies regularization or if regularization is one of many ways in which generalization could be achieved. In other words, are the authors

hypothesizing that generalization is achieved through regularization in the brain? If so, what are the alternatives? Similarly, was L1 norm regularization chosen due to its simplicity or is there something that L1 norm regularization captures that other kinds of regularization do not? It will benefit the reader if these issues are briefly discussed.

Generalization does not entail regularization, but good generalization does. We meant regularization in the most general sense of Wikipedia: *regularization is a process of introducing additional information in order to solve an ill-posed problem or to prevent overfitting*. We imagine that any form of regularization will work well that minimizes the number of neurons used for decision making, including L0 regularization. L1 regularization is often seen as an approximation of L0 regularization, which seems to be the reason why it works well. L2 does not work well in our view because it minimizes the length of the total weight vector, which entails no cost on the number of neurons. (No changes were made to the manuscript with regard to this answer).

Minor points (including typos and comments on clarity):

1. Line 46: ‘hanks’ should be ‘thanks’.

Corrected. (Line 46, page 3)

2. In lines 71-72, the authors say, “The long stimuli S6 to S10 were followed by an air-puff aimed at the experimenter.” Then in lines 76-77, they say, “Either the long stimuli or short stimuli were followed by an air-puff.” The first statement is confusing.

We mean that the air-puff contingency of long and short stimuli was counterbalanced across birds. We correct the figure caption (lines 76-77, page 4) to read “In this example, the long stimuli (S6 – S10) were followed by an air-puff aimed at the experimenter.”

3. Line 104 (fig. 2C): What is the purpose of the smoothing spline fits with parameter 0.3? Are these oscillations significant and should we take them seriously? If not, the solid curves are distracting.

We have increased the smoothness of the spline fits by changing the parameter from 0.3 to 0.00005. Now the distracting (and meaningless) oscillations are gone. Altered Fig 2C on page 6.

4. Line 106 (fig. 2D): Many of the larger solid symbols are lost in the smaller symbols and are barely visible in the plot. Consider changing the plotting scheme for clarity. See also fig. 3B.

We have increased the size of the larger symbols for clarity. Alterations to Fig 2D left panel, page 6 and Fig 3B, page 9.

5. Line 111 (fig. 2E-G): What do the whiskers indicate? See also fig. 3C-D.

We have removed the whiskers (which indicated standard error of the mean) because we now plot the median in Fig. 2E-G (changes on page 6) and Fig. 3C-D (page 9). The range is visible in the raw data points themselves.

6. Line 251: Fig. 4G should be Fig. 4E?

Corrected (now line 259, page 15).

7. Line 310: “the synaptic weights (black)...” should be ...(blue)?

Corrected (now line 319, page 18).

8. Lines 322-323: “...in proportional” should be “...in proportion” or “...proportional”?

Corrected (now line 332, page 19).

9. Line 337: What is alpha?

It is the learning rate of the L1 penalty term λ . Corrected (now line 344, page 19).

10. Lines 487-489: Is it implied that the number of stimulus-puff pairings given to the 3 VL birds is too small for them to learn to escape the puff? This needs to be made clear and explicit.

We presented several hundred stimulus-puff pairings to these 3 VL birds to allow them to learn the aversive nature of the air-puff prior to observing a naïve experimenter. In principle, a single puff might have sufficed to this effect, because single-trial aversive conditioning has been reported in birds.

On the other hand, too many stimulus-puff pairings would be counterproductive because the VL birds would start to learn as do experimenters during the early training phase. We have added this information to the Methods line 503, page 27.

11. Lines 429-430 in main paper and lines 5-6 in sup.: There is a discrepancy in the number of animals used for the study. It says 46 in the main paper and 51 in the supplement.

The actual number of birds is 51 (10 EXP, 9 OBS, 9 GENEXP, 9 GENOBS, 7 PL, 5 VL, 5 –TCOM = 54, but 3 experimenters were also part of the GENEXP group, mentioned in point 1, Bird Groups and Experimental hypothesis, Methods section).

We apologize for this error, the correct number is indeed 51. We added some experiments since the last revision of the manuscript and did not correct these numbers.

Correction made to line 439, page 24 of Methods and line 5, page 1 of Supplementary Information.

12. Supp. Line 6 and main paper lines 62-64: Whereas in the main paper the authors stress that

they only claim to be studying 'observation' learning and not 'observational' learning, they say they are studying 'observational' learning in the supplement. This needs to be consistent.

Corrected (line 6, page 1 of Supplementary Info). We now only use 'observation' learning and not "observational learning".

13. Supp. Line 15: Is 'scuttle bone' meant to be cuttlebone?

Yes, corrected. (*Experimental setup*, line 15 page 1)

14. Supp. Line 77: 'of' should be 'to'?

Corrected. (line 77, page 4).

15: Supp fig 1D: If the y-axis is 'fraction of 8 blocks significant', does it make sense to have data points for the first 7 blocks?

We removed the data points for the first 7 blocks for both EXP and OBS in Fig S1D, page 13 of the Supplementary Information.

16. Supp. Fig. 3B-C: It was not clear why the EXP and OBS are leaving the perch earlier during the un-puffed trials? Please clarify.

EXP and OBS leaved less often during unpuffed trials but on those trials they left earlier on average. We found a uniform distribution of escape responses across trial time in un-puffed trials. However, in puffed trials, we observed an increase in escape rate at some point during the trial time (Fig. S3). Hence, in puffed trials, the distribution of escape events was not uniform but was heavier on the right side. In other words, the average escape time (the center of mass of escape events) was later in puffed trials.

(No changes made)

References:

Cameron, G., Schlund, M. W. and Dymond, S. (2015) Generalization of socially transmitted and instructed avoidance. *Front. Behav. Neurosci.* 9, 159.

Buchanan, J. J. and Wright, D. L. (2011) Generalization of action knowledge following observational learning. *Acta Psychologica* 136, 167-178.

Bird, G. and Heyes, C. (2005) Effector-dependent learning by observation of a finger movement sequence. *Journal of Experimental Psychology: Human Perception and Performance* 31, 262-275

Broeren, S. Lester, K. J., Muris, P. and Field, A. P. (2011) They are afraid of the animal, so therefore I am too: Influence of peer modeling on fear beliefs and approach avoidance

behaviors towards animals in typically developing children. Behaviour Research and Therapy 49, 50-57.

Mineka, S., Davidson, M., Cook, M. and Keir, R. (1984) Observational conditioning of snake fear in rhesus monkeys. Journal of Abnormal Psychology 93, 355-372.

Reviewer #2 (Remarks to the Author):

Comments to "Learning to perform auditory discrimination from observation is efficient but less robust than learning from experience" NCMMS-17-27463

This is an interesting manuscript that shows something that would intuitively be unexpected, that observational learning can be faster than learning from own experience. As the learning mechanism is very basic (conditioning) and the model species is not "too cognitively advanced" (zebra finches) I suspect that this could depend on some basic mechanism. It is also interesting that this quick social learning allows for less generalization compared to learning from own experience. I think that these are important and novel results; however, I also have some problems with the manuscript. One is the volume and detail of the analyses. Many ideas and side-tracks are tested and controlled for, it takes two days to read this manuscript, and to go through all the analyses and ideas in detail would take a week. As I cannot spend that much time on this, I will keep my comments on a general level. If the authors concentrated more on the main findings, I think this would increase readability a lot.

Also, I lack a discussion of why observational learning should be quick but with poor generalization. Is this only a learning trade-off? Is this something typical for zebra finches? Something general? For small finches, couldn't it be important to react quickly to what others are doing? Imagine an airborne predator attacking zebra finches foraging on the ground. A quick response to what others are doing could then be decisive for survival.

Why observation learning is quick but with poor generalization seems to be indeed a learning tradeoff, as the reviewer speculates. Our L1 regularization mechanism provides a detailed illustration of this tradeoff. L1 regularization works by subtracting a fixed amount from each synaptic weight after every weight update, hence it slows down learning. Because observers supposedly are not affected by such subtraction, they learn quickly (and generalize poorly). We now more clearly point out in the discussion that such intuitive tradeoff agrees with L1 regularization implemented as weight subtraction (lines 411-413, page 22).

The discussion of machine learning is very detailed, and it feels somewhat speculative. Maybe you could reduce it?

We cut the paragraph about how to choose the regularization parameter in machine learning and moved it to the Methods Section as a note (moved from page 23 to line 620 on page 32).

Detailed comments:

L 42: what is “sensory learning”? What would “non-sensory learning” be?

We corrected the wording to “sensory discrimination learning” (non-sensory learning could encompass purely motor tasks, or working memory tasks etc) (correction made to line 42, page 2).

L 44: is “cue dependence” the right word? Sounds like you have studied different types of similar cues, for example different types of sound cues.

We have not simply studied the effects of different sound cues, as the sound cues alone presented to PL birds did not elicit observation learning. Therefore, the action of the airpuff against the body of the experimenter was a key somatosensory cue. This cue was absent to observers, thus we believe that “cue” is the appropriate word. (No changes have been made to the manuscript)

L 46: You are probably correct in that zebra finches are useful models, but their ability to detect subtle sound differences is probably not better than other passerines. Maybe you should add that they are easy to breed in captivity.

We added that zebra finches are easy to breed to the Supplementary information line 8. We also provide a brief review of zebra finch auditory perceptual capabilities in the Introduction section (lines 46-54 page 3).

L 50: Don’t you think that they already could discriminate the sounds and that you trained them how to react to them?

This is possible indeed, we cannot easily distinguish what birds are able to do from what they want to do. We revised the introduction and now write “we trained one of the two birds in a pair to behaviorally discriminate short from long renditions of a zebra finch song syllable” (line 55s, page 3), to leave it open as to whether birds learned the sensory discrimination or the behavioral response.

L 51: Zebra finch sounds are probably the most appropriate ones for a study such as this. Do you think that there could be any “bias” in them, for example that the finches could be genetically pre conditioned to react to them in some way?

There are indeed several studies showing such an innate bias, i.e. zebra finches prefer to learn zebra finch songs if given a choice between zebra finch and Bengalese finch songs. However, any genetic bias will be equally affect experimenters and observers and so cannot account for differences between them. (No alterations made to the manuscript).

L. 55. Did the experimenter finches ever give any alarm or aversive calls at the puff that could work as non-controlled cues?

We have observed alarm calls within trials on rare occasions, most of the time we see contact calls (like Tet, Stack or Tuk calls). However, quantifying how much of which type of call is present in the data would require a considerable (and manual) effort to cluster and count the call types, which we find unnecessary since the –TCOM group exactly controls for the effect of vocal communication on observation learning. (No alterations made to the manuscript).

L. 68: These measures are also important outside the machine learning community!
Figure 1. The most vital part of the figure is C) do you really need the rest? It is also explained in the text.

To maintain high readability, we prefer to keep all the panels in Fig. 1. Fig 1B shows all our stimuli, so that readers can verify their overall similarity (and minute differences) between training and generalization sets of stimuli. (No alterations made to the manuscript).

L. 71-72 and 76-77: Why did you consistently use the long sound in association with the air puff and either the long or short in c)?

This point was raised by reviewer 1 as well and we corrected the caption of Fig. 1. We counterbalanced the contingency of the puff to long and short stimulus classes.

L. 81: Why is this pre-training rather than different types of training?

In the monkey research literature, such familiarization with the setup is usually referred to as pre-training. (No alterations made to the manuscript).

Fig 2. This figure has very much information in it. Is everything necessary, for example a) and b)?

The figure is only about difference in escape probability (dPesc), so has quite a narrow focus in our view. Panels a and b illustrate the manner in which dPesc is computed and how it behaves in a single bird, so we believe this is ‘raw data’ needed to develop an intuition about our measurements. (No alterations made to the manuscript).

L. 96: Why were they separated by a screen? Could the observers see the reaction of the experimenter at the puff?

Usually experimenters are blown off the perch when hit by a puff. Any (visual) reaction away from the perch is not visible to the observer. The screen is necessary to attract the experimenters back onto the perch, which is the sole location from which birds can see each other. Birds can freely exchange vocal information at all times (in all groups except –TCOM).

(No alterations made to the manuscript).

L. 119: You mean one-sided? How can you use a directional hypothesis for something that was not beforehand expected? Even if hypotheses are directional, and the opposite outcome doesn't make sense, most journals want two-sided tests. I would have expected that own experience would give faster learning. Also, overall, even if you calculate effect sizes and make power analyses, sample sizes are really small.

We expected observers to be faster. Similar to experimenters, they received a pre-training phase. However, unlike experimenters, they have gotten an additional observation phase. So overall, they had more experience with the stimuli than experimenters and so we expected them to learn more rapidly. We use very conservative non-parametric tests for between-group comparisons (we make no Gaussian distribution assumption). Second, our effect sizes are "large". Finally, we also performed two-tailed tests, and our results are upheld (see Supplementary section *Robustness of statistics*, point 3.) (No alterations made to the manuscript).

L. 132: How can they be exposed to a pre-training phase when they have already been trained to be observers? Why was this pre-training necessary?

Pre training was necessary to give observers and experimenters the same familiarization with the setup. We wanted to exclude any confounding explanation for why observers behave differently than experimenters when processing the full set of sensory stimuli. (No alterations made to the manuscript).

L. 136-140. Is the test statistic T from Wilcoxon? You have a dispersion measure at the mean but also give 95% CI separately. Why? Are they different?

Yes, the test-statistic is the rank statistic from a Wilcoxon rank sum test. What matters in this test is the median, not the mean. We therefore removed the effect size measure Cohen's d because the latter is computed with respect to the mean. We have changed the reported location and dispersion measures to median and range and moved the Cohen's d effect sizes to the Supplementary material (section *Robustness of statistics* point 6, page 12). The 95% confidence interval is not related to a dispersion measure on the data; it is an interval which would contain (with 95% probability) the observed difference in medians between GENEXP and GENOBS (for dPesc and Trials to criterion).

Median and range now reported on line 94-96, page 5; Fig 2 E,F and G, page 6; lines 116, 123, 127 on page 7; line 141, 144, 148 on page 8; Fig 3C and D on page 9; line 165 page 9; lines 185, 186, 190 on page 10; line 190 on page 11; Fig 4E on page 12; lines 209, 219, 220, 223 on page 13; lines 241, 242, 244 on page 14.

L. L. 141: Is this one- or two-sided? Why don't you state this here as you do above?

We corrected “two-tailed” to “two-sided”. We have fixed this at other places in the text where a statistical test is performed.

Changes made to line 128 page 7, line 142, 145, 148 and 151 on page 8, line 191 and 193 on page 11, line 224 on page 13, line 226, 242, 245 on page 14, line 248 and 259 on page 15, lines 486 and 488 on page 26.

L. 161: What do you mean by similar here? How did you compare them? Video analyses?

The subsequent sentence in the manuscript provides the requested information: *they tended to abruptly increase their perch escape rates just before air-puff onsets, Fig. S3A, B.* Hence, we did not use video analysis, but simply quantified escape timing. (No changes made to the manuscript).

L. 165: What would a passive perceptual process be here?

By passive we mean that the observers could have gained an advantage in learning because they could have learned a) a good (useful) stimulus representation and b) to discriminate these stimuli while ignoring the behavior of the experimenter (hence, the sound of the air-puff could have served as a supervisory label to discriminate the two classes of stimuli). (No changes to the manuscript).

L. 169. Several thousand? Don't you re-condition their response by this?

To implement a good control for perceptual learning, we needed to use as many stimulus playbacks as used in the original EXP-OBS experiment. If we were to play back many fewer stimuli, then one could argue that birds start learning passively only after several thousand playbacks. Thus, it was unavoidable to use several thousand playbacks in this control experiment. (No changes to the manuscript).

L. 178-181: How can you calculate a dispersion measure but not 95% confidence intervals?

Again, the dispersion measure is standard deviation. We have now replaced the location and dispersion measure (previously mean and standard deviation) with median and range since this corresponds to the non-parametric test we use (Wilcoxon ranksum). The 95% confidence interval cannot be computed if there are ties in the data. That is also why the p-value is not exact. The 95% confidence interval is an interval for the estimate of the difference in medians, not the dispersion of the data (Trials to criterion or dPesc). (Same changes as question above regarding L 136-140).

L 191: What does this sentence mean? “Observers learn from behaving, expert experimenters,..”. It does not make sense to me.

We changed the sentence to “Observers learn from behaving experts” (now line 197, page 12). We mean to say that the experimenters must be engaged in the task (unlike the PL experimenters) and must be able to discriminate the stimuli with high accuracy (unlike the VL experimenters).

L. 192-193: Could the birds themselves trigger this?

Yes, both PL and VL birds trigger stimulus and air-puff presentations by sitting on the perch just like a normal experimenter. The difference is that a) they cannot physically experience the air-puff (PL experimenter) and b) they are not given enough time to learn (VL experimenter). (No changes made to manuscript)

L. 204: “Expert” models? This sounds like there were some really good ones among the ones that learnt. Would the ones that did not learn induce any learning at all? I guess they did if they were closer to EXP than OBS?

“Expert models” refers to experimenters that reached our performance criterion. All birds in the EXP group reached the performance criterion. So all EXP became experts. On the other hand, VL experimenters did not reach the performance criterion (and VL observers did not learn quickly, Fig. 4). (No changes made to manuscript)

L 229-230: Rather than “speculated”, did you not actually want to control that it did not?

We first speculated that vocal exchanges might be important and then controlled for them using the –TCOM group (Fig. 4). (No changes made to manuscript)

L. 229-233: What type of calling? Alarm calls?

We observed mostly “Tet” and “Stack” types of calls, which are normal contact calls of zebra finches. We observed few Alarm calls but we have not quantified the number of each type. (No changes made to manuscript)

L. 239: Why do you use a parametric test here, but otherwise non-parametric? Was the t-test two-sided?

In order to avoid confusion, we have reverted to non-parametric tests for all the statistical tests we perform. We now perform Wilcoxon signed rank and rank sum tests for comparisons of median calling rates in puffed and unpuffed trials (lines 240-248, page 14). However, we did not see any difference in our results when changing from parametric to non-parametric tests.

L. 245-246. This is a nice design!

Thanks.

L. 254: Would anyone claim this? Do you really need to test for it?

We state this because it is the consequence of the –TCOM experiment. (No changes made to manuscript)

L. 287: I would suggest that both own experience and observation gives experience.

We agree and corrected the sentence. We now write “... these theoretical insights from statistical learning theory suggest that direct experience of the learning cue (here, reward/punishment) is associated with regularization whereas observation is not.” (Change made to line 296 on page 17).

L. 271-352: I would be cautious with making too strong inferences from machine learning. I think the authors have a point here, and that comparisons can give us insights. But machine learning is just machine learning and we don't know to what degree it actually resembles what goes on in real bird brains. When machine learning is used to model learning in animals it will in most cases be a simulation of learning, not exactly the same process. I am not at all sure that the machine learning dilemma (too specific but rapid learning vs slower but more useful general learning) should be directly applicable to what we see in the zebra finches.

We agree that caution is advised in general when modeling. The analogy with machine learning may not be the only explanation of our findings, but it is the simplest we could find. We tried to explain behavior with the principles of a formal theory, and find that simple, biologically plausible learning (our learning rule uses only quantities local to a synapse) can reproduce behavior. Of course, a machine learning technique is not a direct description of neural processes, but in many cases theoretical models of neural activity are abstract but useful (e.g. L1 regularization based receptive field estimation or sparse coding theory, Olshausen and Field 1996).

We believe that the literature on animal behavior and human cognition has not widely adopted or sufficiently utilized the principles of formal statistical learning theory. Prior to our manuscript, there may not have been much reason to assume that rapid & poor learning opposes slow & good learning. However, based on our observations, we believe that the Bias-Variance dilemma is not only a principle applicable to artificial learning systems but also to natural systems. We hope such conjecture can guide future experiments on this topic. (No changes made to the manuscript).

It could be something like that, but it could also be something completely different. In what way are zebra finches evolved to react on social learning cues? By the way, do you really need this detailed machine-learning section?

We prefer to keep the discussion of machine learning because it helps motivate the analogy between experimenter-observer behavior and model complexity and overfitting. We removed the technical parts of the discussion into the methods. (Lines 417-424 on page 23 in Discussion moved to Methods line 621-627 on page 32 and 33).

L. 305-306. Is this because more noise makes generalization harder?

Yes, because if extra uninformative (noisy) features are added to the stimuli, generalization performance is bound to decrease (because some noisy features will by chance be informative on the training set but not on the generalization set – because they are noise). (No changes made to manuscript).

L. 319-321: This does not say anything to me. Could you explain it in a simpler way? The same is true for L. 345-347.

By adding noise to the labels (i.e. whether a stimulus is puffed or unpuffed) during the simulation of our learning rule, we essentially simulate that observers get noisy information from experimenters (since none of the experimenters was 100% correct). We find that the simulation results are robust to this label noise: the curves are similar, no matter whether the label is the air-puff sound (which indicates the label of a stimulus with complete certainty) or the actions of the experimenter (which provides uncertain information about the label of a stimulus). Therefore, we cannot determine which source of information is used by the observer, at least under the model assumptions. (No changes made to manuscript).

L. 355: Could they not discriminate these stimuli already before the experiment? Do you not mean “learn to react to auditory stimuli...”

We rephrased the sentence to ‘can learn to discriminatively respond to auditory stimuli’ (line 369, page 22). However, it must be emphasized that for a given discrimination paradigm, different stimuli can require different amounts of time to be discriminated (Canopoli et al. 2014 and work in preparation). So if the question of learning is only about learning to react to stimuli that can already be readily discriminated, then different zebra finch songs should be discriminated well within (roughly) the same training time. So we prefer the interpretation that birds could not already discriminate the stimuli.

L. 356: have you actually demonstrated this trade-off? Your results suggest it, but that is not the same thing. What was the trade-off, learning speed vs generalization ability?

Yes, we find a trade-off between learning speed and generalization ability. This trade-off is clearly demonstrated in Figures 2 and 3. (No change made to the manuscript).

L. 361-362: Why would this be so? Is it only a trade-off depending on the learning type or could the birds have natural, evolved responses that differ depending on ecological factors?

We are not sure about the reasons of this tradeoff, but we suspect that it is a combination of both. There may be an evolutionary pressure to efficiently deal with this tradeoff (to experience is expensive and dangerous but to observe is cheap and safe). (No change made to the manuscript).

L. 366-367. Yes, in this system. If you look at a more cognitively advanced bird, such as a raven, social learning is probably different, occurring on a high cognitive level with lots of generalization (like you say on lines 370-371).

We agree. (No change made to the manuscript).

L.395: How did the amygdala come into this? As I know it, it is a site for emotional memory.

We use this to point out a case where subtractive weight normalization is experimentally confirmed to support our modelling choice of L1 regularization. (No change made to the manuscript).

L. 405-406. Well, maybe you should not discuss it in such detail then.

We cut lines 283-288 on page 16 in the Results section which introduce the analogy with overfitting and underfitting in machine learning and we moved the sentences regarding techniques of regularization in machine learning from the Discussion to the Methods (Lines 417-424 on page 23).

L. 410: Another possibility could be that it has nothing to do with regularisation at all.

We mean 'regularization' in the most general sense (see also our reply to reviewer 1).

L. 470: What is a biological replicate?

A biological replicate is another sample of the same population undergoing the same measurement or experimental test. (No change made to the manuscript).

L. 539: I would say, in case of negative results, power tests are always useful. Also, $n = 9$ is still a very small sample size. Even if you use a non-parametric test a larger sample size would have been good. It is hard to motivate a sample size smaller than 10 with statistical reasons.

We have designed our experiments with this sample size in mind, and so to add more birds at this point in time would obscure reported p values because of cumulative testing. Most studies on zebra finches use sample sizes of $n=6$, so we are well above this threshold. (No change made to the manuscript).

Reviewer #3 (Remarks to the Author):

This study aims to study the mechanisms of observational learning in birds (zebra finches). The ms is composed of 2 parts: an experimental part and a modeling part.

The authors used an operant conditioning task with mild punishment (air puff). During the test phase, the birds were trained to discriminate sound stimuli ($n = 10$) classified into 2 categories. These sound stimuli were constructed from multiple copies of a song syllable of a male finch. The authors selected 10 copies of different durations and prepared each stimulus by repeating the syllable 6 times with constant inter-syllabic intervals. They created 2 categories of stimuli: short and long. During a test phase, the bird had to learn to discriminate the sounds of the 2 categories. During the generalization phase, the bird had to learn to discriminate new stimuli. These new stimuli were constructed in the same way as those generated for the test phase, but using exemplars of the song syllable of the same male recorded the next day.

In a first experiment, a female observer (OBS) observed the tested female (EXP) during the learning phase, before being tested in the presence of a new female naive observer. The authors observed that OBS females learned the task of auditory discrimination faster than EXP females.

In a second experiment, a female observer (GENOBS) observed the tested female (GENEXP) during the learning phase. The two females were then placed each with a new female naive observer. In this situation, the authors observed that GENOBS females took longer to learn to discriminate the set of stimuli used for generalization. The authors interpret this result by the fact that observers (OBS) react to small acoustic differences between the stimuli used for training and generalization whether EXP birds focused more on the duration of the stimuli.

In a third experiment, the authors used the same paradigm as in the first experiment but they directed the airflow so that it did not affect the tested female (EXP) while another female watched her (PL). Thus, only the sound of the air puff was audible during the learning phase. Not surprisingly, the bird EXP did not learn to discriminate the stimuli of the 2 categories. When the PL birds were tested with the airflow directed at them, they took longer than the OBS birds to discriminate the stimuli broadcast during the training phase. This difference suggests that OBS learned the discrimination task by observing the behavioral response of the EXPs during the air puff, and not by being passively exposed to the noise of the air puff.

After observing rapid learning in OBS and slow learning in PL, the authors questioned whether experimenters should be experts to induce learning in observers. They exposed 5 females (VL) to experimenters who had not yet achieved the task of discrimination. They also exposed 3 VL to air puffs (about 500) following the same protocol used for the first experiment, before giving them the opportunity to observe experimenters who had not yet succeeded in the task of discrimination. The authors observed that the performance of the VL birds was weak suggesting that the OBS did not learn by predicting the value of the reward and by converting this prediction into optimal action. On the contrary, the behavior of the VLs indicates that the OBS focused their attention on discriminative actions of experimenters.

In another experiment, the authors investigated the influence of vocal communication on learning. They observed a change in vocal activity at the end of the training phase between

puffed and unpuffed trials. Thus, vocal activity might facilitate learning. The authors used the same procedure as in experiment#1 but they placed the experimenter and the observer in two separate cages. This setup permitted to control the vocal exchanges that were broadcast by a loudspeaker in each cage. The authors prevented the vocal exchanges between the beginning of the broadcast of the sound stimulus and the noise of the air puff. The tested birds (-T-COM) exhibited learning performances comparable to those of OBS of the 1st experiment suggesting that learning was not facilitated by the vocal interactions during the broadcast of the sound stimuli.

In a final experiment, the authors checked whether observational learning was not a simple form of stimulus enhancement. To do this, they modulated the power of the air puff during the pre-testing trials and observed that a weak air puff linked to a sound stimulus did not induce escape behavior in OBS. This means that in addition to the knowledge of the stimuli and the action of the experimenters during the broadcast of these stimuli, the observers also need to make the aversive experience with the air puffs to express their learned knowledge.

In a second part, the authors relied on knowledge in machine learning and statistical learning to propose a simple mechanism that could explain these experimental results. This mechanism is based on a tradeoff between speed of learning and generalization. They used an artificial neuron that received input from a group of at least 22 neurons tuned to various acoustic parameters. This simulation showed that the weak generalization observed in OBS could be explained by taking into account too many acoustic characteristics of the stimuli broadcast during training. Thus, in the neurons observers, the weight of the various acoustic parameters in the classification is balanced whereas for the experimental neurons, the classification is based on the duration of the stimulus which is the criterion for discrimination between the two categories.

Then, the authors manipulated another parameter (regularization parameter) to integrate into their model another aspect linked to the activation of reward circuits during learning (dopaminergic neurons). The idea is that the observers' brain would not modulate this parameter because observers do not directly experience a punishment during the experimenter training phase. From these simulations, the authors conclude that this regularization parameter should be taken into account to obtain a good agreement with the experimental results, and in particular in the observed differences between the performance of observers and experimenters. Using their model, the authors did not succeed in distinguishing between the two possible learning strategies of the observers based on: 1) air puff's auditory cue (the actions of the experimenter directed to the air puff) or 2) the escape behavior of the experimenter.

In the discussion, the authors conclude that an experienced cue favors robust generalization. They also discuss the results of their study in relation to the conceptual and theoretical framework of social learning in animals. They also offer explanations and future directions for exploring the behavioral and neurobiological substrates of the results obtained.

This is a very elegant study, carefully dissecting the behavioral mechanisms at stake during observational learning in animals. The modeling part nicely complements the experimental part. The statistical treatment seems correct and the data justify the conclusion. This approach is original and the results of this study will be of interest for researchers interested in mechanisms of learning from different fields such as animal behavior, computational

neuroscience and evolutionary robotics.

My only concerns are:

1) The use of a single sex (female) in a species for which it has been demonstrated that males and females do not deal with social information in the same way. See for example the study by Katz and Lachlan (Animal Cognition, 2003) showing that social learning of food types in zebra finches is directed by demonstrator sex.

We agree and added a sentence to the discussion (lines 384-386 page 21), citing the Katz and Lachlan paper and also a recent paper by Tokarev et al.

2) The experimental procedure involves a mild punishment (air puff); not a reward or a stronger punishment (electric shock). The authors should clarify whether their model takes into account those aspects that may affect learning.

Indeed, neither our model nor our study really addresses dependence of our findings on reinforcer type, we added a sentence to this effect to the discussion (lines 386-387 page 21). Our only observation in that respect is that for mild puffs during pre-training, we did not see good performance in experimenters, which we used to rule out simple stimulus enhancement. These findings also show that a minimum air puff strength is necessary to trigger stimulus-dependent escape behaviors. The effects of more aversive stimuli (e.g. electroshocks) would need to be investigated.

Here are some more specific comments:

Line 91-92. Was the observer systematically moved to the experimental chamber of the experimenter she observed (same location)? Or is it possible that an observer stayed in her part of the cage (observed chamber) or was moved to another cage of another setup? This change of location seems to be the case for the GENEXP/GENOBS (lines 130-131) and could affect the learning. Please comment.

EXP and OBS were in the same experimental chamber, but in adjacent cages (a chamber is a large acoustically isolated box that can house two cages). When the EXP finished training, the OBS was moved to the EXP cage, a naïve observer was added to the OBS cage and the EXP was removed.

GENEXP birds were moved to another chamber (a new cage with a naïve observer in the adjacent cage) while the GENOBS bird was again simply moved to the old cage of the GENEXP bird. So all in all, GENOBS and OBS are not different in that sense, only GENEXP birds were moved to an entirely new chamber. We added this information to the Methods Section (line 482 page 26).

We made sure that all experimental chambers, cages and food and water locations were consistent for all birds.

Line 178. The authors could also compare the results with EXP. They provide this comparison in line 221 but it could be mentioned here.

We moved the comparison from line 221 to line 178 (now line 193, page 11) as suggested.

Line 210. It is not clear whether 3 additional VLs were initially exposed to air puffs or if these 3 VLs are part of the group of the 5VLs. Please comment. See also line 135 of the supplementary information.

These 3 birds are part of the 5 VL birds, we clarified this point in the Methods Section (line 506, page 27).

Line 232. How could the authors distinguish the vocal activity of OBS from EXP? Was calling rate inspected in OBS (as mentioned in the text) or in EXP? I guess there is a mistake in the text (mention of OBS but not EXP). Please comment.

We did not try to distinguish EXP from OBS vocalizations. We corrected the text and replaced 'OBS' by 'EXP-OBS pairs' (lines 243 and 248 page 14).

Lines 327-328. The authors mention the possibility that birds could experience a reward but using this experimental procedure, it is a mild punishment. Please comment.

We agree. The possibility of using rewards or strong punishments is now mentioned elsewhere in the discussion (line 387, page 21).

Lines 381-382. To de-emphasize the importance of vocal communication, the authors should have checked whether auditory cues alone (vocal but not visual communication allowed) could trigger training in OBS. There are also evidence that stress can be coded in animal vocalizations including calls in finches (see Perez et al. Horm. Behav. 2016). There are possibilities that calls produced following a stimulus linked to an air puff might bear information about stress or discomfort.

We did not do this test, such an experiment could have been done indeed. However, in our manuscript, we were interested in whether vocalizations are necessary and less so whether vocalizations are sufficient. (No changes made to manuscript).

Line 412. Please correct: neuromodulatory neurons.

Corrected (now line 424 page 22).

Supplementary information, line 7. Male zebra finches do not defend a territory both in the field and in the artificial conditions of the laboratory. This statement is not correct. Male finches

produce songs in the courtship context and probably also to facilitate group cohesion. Also, male is the predominantly used sex for neurobiological studies involving song learning since females do not sing. But both male and female finches are used in labs investigating other aspects of the social life.

We wrote ‘females choose their mates by listening to the songs of competing males in their territory’ to mean that males compete for females. To make this point clearer we removed ‘in their territory’ (line 7, page 1 of supplementary).

Supplementary information, line 51. The authors should provide measures of acoustic parameters of these different renditions of the same song syllable that was used for these experiments. They should also provide a short review of the literature regarding the capacities of auditory discrimination in both time and frequency domains in finches. This would help the reader to estimate whether finches are able to discriminate two sounds with a high acoustic similarity such as two different renditions of a same song syllable produced by a single individual.

We added a figure to the supplement in the Supplement (Fig S5, page 18), showing the correlation coefficient between duration and the other 21 acoustic features used to represent stimuli in our simulations.

With regard to the discussion of the work on fine discrimination of acoustic signals by zebra finches, we integrated the following information into the introduction (line 48 to 55, page 3):

The auditory perceptual capabilities of zebra finches are well documented in tasks such as temporal integration of pure tones (Dooling 1979) and discrimination of complex stimuli such as contact calls through cues such as timbre and harmonicity (Lohr and Dooling 1998; Cynx et al 1990; Cynx 1993). Zebra finches are even sensitive to variations in human speech (Ohms et al. 2010; Spierings and ten Cate 2014). Recently, Lohr et al (2006) showed that zebra finches can accurately detect changes in the fine temporal structure of contact calls, changes occurring within 1.2 ms.

Discussion. The authors should also discuss about the possibilities that being observed might affect the speed of learning. They took care about it by always having two birds in the experimental setup but there is a possibility that such eavesdropping might affect the bird's behavior. Please comment.

Indeed, the presence of an observer might have affected learning in EXP, we added a sentence to the discussion (line 387 page 21).

Currently, we believe that the presence of the observer as a social reinforcer is needed for EXP to keep going back to the air-puff perch. Hence, it is currently not possible to quantify behavior in singly housed EXP.

Methods, escape behavior. The authors should provide data about the escape time. Did the trained/tested bird systematically wait for the end of the stimulus broadcast to escape in case of punishment?

We provided a detailed analysis of escape behavior in Supp. Fig. S3. (No changes made to the manuscript).

REVIEWERS' COMMENTS:

Reviewer #1 (Remarks to the Author):

The authors have provided acceptable responses to most of the concerns expressed in Round 1 of the review. However, there remain a couple of areas where the manuscript can be improved, as listed below:

Major point 3: The issue of whether generalization automatically implies regularization was brought up by Reviewer 2 as well. This question is likely to arise in readers of this manuscript (especially those not very familiar with machine learning), as is the rationale for using L1 regularization. The authors' response to these questions sounds reasonable but does nothing to improve the manuscript if it is not included somewhere in the manuscript. The manuscript will greatly benefit by a brief discussion of a) what the authors mean by regularization and b) why focus on L1 regularization. Even including a brief discussion like the authors present in their response to reviewers might be sufficient.

Minor point 10: The key missing detail here was that "...too many stimulus-puff pairings would be counterproductive because VL birds would start to learn as do experimenters..." It will improve the clarity of the presentation if this rationale is explicitly stated in the Methods section (lines following 503).

Reviewer #2 (Remarks to the Author):

I have now reread the manuscript after the changes. I think that most changes and answers to my and the other referees' comments are satisfactory. I still think that in many places is a bit too complex and that I would facilitate for readers to concentrate on the main messages. This, however, is up to the editor, maybe it is the style in Nature communications?

I have a few minor comments to the authors' answers to my previous comments:

L. 68 Well, the figure may become more complete with all this detail, but I still think that many subplots will decrease readability rather than increase it.

Figure 2: yes, but it is a very "broad" figure for having a narrow focus.

L. 119: Yes, but even if you have a directional hypothesis one-sided stats can only be used if the opposite 2.5% are scientifically meaningless. What would you have done if observers had been significantly slower? Refrained from interpreting it?

Reviewer #3 (Remarks to the Author):

The authors have carried out a satisfactory revision. They took into account all my comments. Please find below some comments for a minor revision.

Page 13, line 188. Since the authors changed to median and range in the revised version, it would be welcome to present these values instead of mean and standard deviation in the whole text. See also legend of figure S4.

Figure S5. Since this figure is not cited in the text, the authors should give more information about these results in the supplementary or in the legend. I understood that about 20 acoustic features have been measured using the Sound Analysis Pro software. The authors should provide this list of features either as a table or in the legend. In addition, they should comment the results reported in this figure, in particular for features showing a significant correlation with duration.

We thank all reviewers for their very helpful comments that allowed us to improve our manuscript.

REVIEWERS' COMMENTS:

Reviewer #1 (Remarks to the Author):

The authors have provided acceptable responses to most of the concerns expressed in Round 1 of the review. However, there remain a couple of areas where the manuscript can be improved, as listed below:

Major point 3: The issue of whether generalization automatically implies regularization was brought up by Reviewer 2 as well. This question is likely to arise in readers of this manuscript (especially those not very familiar with machine learning), as is the rationale for using L1 regularization. The authors' response to these questions sounds reasonable but does nothing to improve the manuscript if it is not included somewhere in the manuscript. The manuscript will greatly benefit by a brief discussion of a) what the authors mean by regularization and b) why focus on L1 regularization. Even including a brief discussion like the authors present in their response to reviewers might be sufficient.

We understand that the concept (and terminology) of regularization may be confusing and unknown to the general reader. We have revised our discussion to clarify what we mean by regularization (...regularization, which is an umbrella term for all kinds of processes that prevent overfitting by introducing additional information, such as for example to use as few nonzero parameters as possible during fitting. Essentially, regularization methods improve generalization performance by dynamically regulating the use of parameters and of training data.)

To explain L1 regularization, we write: *L1 regularization implements a conjunctive minimization of summed absolute synaptic weights¹⁸ that we implemented on each synaptic weight update as a small reduction by an amount λ ²³. We focused on L1 regularization, because it allowed us to formulate a simple online model that sets the regularization parameter λ as a function of past experienced reward. We write: *We used L1 regularization because it is very simple (subtractive) and because it allowed us to formulate a mechanism that dynamically regulates the regularization parameter λ in proportion to reward prediction error (Methods), known to be signaled in the vertebrate brain by a class of dopaminergic neurons.**

Minor point 10: The key missing detail here was that "...too many stimulus-puff pairings would be counterproductive because VL birds would start to learn as do experimenters..." It will improve the clarity of the presentation if this rationale is explicitly stated in the Methods section (lines following 503).

Corrected: We have included this statement after line 507, page 27 in the same paragraph in the Methods section.

Reviewer #2 (Remarks to the Author):

I have now reread the manuscript after the changes. I think that most changes and answers to my and the other referees' comments are satisfactory. I still think that in many places is a bit too complex and that I would facilitate for readers to concentrate on the main messages. This, however, is up to the editor, maybe it is the style in Nature communications?

I have a few minor comments to the authors' answers to my previous comments:

L. 68 Well, the figure may become more complete with all this detail, but I still think that many subplots will decrease readability rather than increase it. Figure 2: yes, but it is a very "broad" figure for having a narrow focus.

We have removed panel 2d from the figure to reduce the number of subplots and simplify the Figure. Figure 1 has just 3 panels A, B, and C, which we believe does not overburden the reader and achieves high readability.

L. 119: Yes, but even if you have a directional hypothesis one-sided stats can only be used if the opposite 2.5% are scientifically meaningless. What would you have done if observers had been significantly slower? Refrained from interpreting it?

We have changed the text and always use a two-sided test now. Essentially, our result still stands because a two-tailed p-value is simply 2 x one-tailed p-value. Since our p-values are small (< 0.01 for all one-tailed tests, and Cohen's d effect sizes are large), there is no qualitative statistical difference except for a slight reduction in power.

Reviewer #3 (Remarks to the Author):

The authors have carried out a satisfactory revision. They took into account all my comments. Please find below some comments for a minor revision.

Page 13, line 188. Since the authors changed to median and range in the revised version, it would be welcome to present these values instead of mean and standard deviation in the whole text. See also legend of figure S4.

We have made the necessary changes to the manuscript and the Supplementary figure S4.

Figure S5. Since this figure is not cited in the text, the authors should give more information about these results in the supplementary or in the legend. I understood that about 20 acoustic features have been measured using the Sound Analysis Pro software. The authors should provide this list of features either

as a table or in the legend. In addition, they should comment the results reported in this figure, in particular for features showing a significant correlation with duration.

We now cite Figure S5 (now Figure S4) in the main text.

We have included a Table (S1) in the Supplementary Information that lists the name of each SAP acoustic feature, a brief description, and the Pearson correlation coefficient (including p-value) with duration (the 21st feature). Features that correlate highly with duration are Mean Frequency Modulation (averaged over the length of the syllable), Wiener entropy, Variance of Amplitude Modulation, Mean Pitch Goodness, and Variance of Pitch Goodness, and Median pitch in diverse time windows within the syllable. Thus, these features are all informative about stimulus class, explaining why in simulations, observer neurons tend to form nonzero synaptic weights with these inputs.